# Modeling treatment and temperature effects on dengue transmission at the division level in Bangladesh

**Md Mafizer Rahman[1], Haridas K. Das[2,3]\*, Sazia Khatun Tithi[1], Md Tonmoy Ul Hasan[1], Moyuri Khatun[1], Md Abdul Kuddus[1]\***

1 Department of Mathematics, University of Rajshahi, Rajshahi, Bangladesh, 2 Department of Mathematics, Oklahoma State University, Stillwater, Oklahoma, United States of America, 3 Department of Mathematics, University of Dhaka, Dhaka, Bangladesh

\* makuddus.math@ru.ac.bd (MAK); haridas.das@okstate.edu (HKD)

## Abstract

Dengue fever remains a growing public health threat in Bangladesh, with urbanization, temperature variability, and limited healthcare resources exacerbating recurrent outbreaks. Although many studies have modeled dengue dynamics, the explicit role of treatment under temperature variability remains poorly quantified. Here, we present a simple and interpretable Susceptible–Infected–Treated–Recovered–Susceptible (SITRS) model for humans, coupled with a mosquito Susceptible–Infected (SI) model incorporating a temperature-dependent biting rate. This framework captures how treatment access and efficacy interact non linearly with temperature-driven changes in mosquito biting behavior. Unlike typical dengue models that assume homogeneous recovery, our formulation distinguishes natural recovery from supportive care, reflecting healthcare disparities between urban and rural regions. Using Lyapunov stability theory, we establish threshold conditions for endemicity. We calibrate the model using division-level dengue surveillance and temperature data across Bangladesh. The results show that limited treatment access substantially amplifies outbreak peaks, whereas timely supportive care reduces epidemic intensity even under high-transmission conditions. Short-term forecasts for 2025 identify Dhaka Metropolitan as both treatment-sensitive and a hotspot, highlighting significant regional inequities in transmission risk. Beyond Bangladesh, this modeling framework offers a generalizable approach for integrating treatment capacity with temperature-sensitive vector dynamics, providing actionable insights for epidemic preparedness in resource-limited settings.

## 1. Introduction

Dengue is a vector-borne viral disease transmitted primarily by *Aedes aegypti* and *Aedes albopictus* mosquitoes, affecting tropical and subtropical regions of Asia and

the Creative Commons CC0 public domain dedication.

**Data availability statement:** All data used in this study are accessible from the publicly available website: https://old.dghs.gov.bd/index.php/bd/home/5200-daily-dengue-status-report.

**Funding:** The author(s) received no specific funding for this work.

**Competing interests:** The authors have declared that no competing interests exist.

Latin America. These vectors also transmit other pathogens, such as Chikungunya and Zika, posing a multifaceted public health threat. Dengue is caused by four distinct serotypes: DENV-1, DENV-2, DENV-3, and DENV-4 [1]. Infection with one serotype confers immunity against that serotype but leaves individuals susceptible to others, allowing multiple infections over a lifetime [2]. Transmission occurs mainly through mosquito bites during the day, although night-time biting can occur, and mosquitoes often feed on multiple hosts in quick succession, facilitating rapid disease spread. The virus persists in the human bloodstream for two to seven days [3], coinciding with the symptomatic period, enabling uninfected mosquitoes to acquire and transmit the virus. Aedes mosquitoes breed in stagnant water found in domestic containers such as pots, buckets, and discarded tires [4–7].

Globally, dengue imposes a heavy public health burden, with an estimated 400 million infections annually [8], of which about 100 million develop clinical symptoms, and approximately 40,000 deaths occur each year from severe dengue. Nearly half of the world's population is at risk, and the disease is endemic in 129 countries, with Asia accounting for roughly 70% of cases. Between 1990 and 2021, reported cases increased by 88% [9], though many go unreported due to asymptomatic infections, self-management, or misdiagnosis. In 2023 alone, over six million cases and 6,000 deaths were reported globally, and by January 2024, more than 500 new cases and 100 deaths had already been recorded [10–12]. In addition, the biting rate generally increases with temperature up to an optimal range, as mosquito metabolism and feeding activity accelerate, but declines at higher temperatures [13].

Amid this global concern, Bangladesh represents a critical hotspot, yet limited attention has been given to dengue transmission modeling tailored to its tropical environment. Most importantly, dengue fever is a major public health threat in Bangladesh, especially in Dhaka city, where it has remained endemic for decades. Historically called "Dacca fever," it was first reported in the 1960s and has since become a recurring public health challenge due to expanding *Aedes* habitats and rapid urbanization [14]. In 2023, Bangladesh faced its worst epidemic to date, with more than 321,000 cases and 1,700 deaths. In 2024, cases declined to 101,214 with 575 deaths [15], while 35,471 cases were reported by September 9, 2025. More specifically, annual outbreaks since 2000 have intensified in scale and severity, highlighting the urgent need for realistic, targeted intervention strategies tailored to Bangladesh's epidemiological and environmental context.

Mathematical modeling plays a central role in understanding transmission and informing control [16]. Classic compartmental models, including SIR and its extensions (e.g., SEIR, SEIRS), have been employed to capture transmission dynamics and seasonal trends [17–21]. These models often incorporate climatic drivers such as humidity and temperature to simulate mosquito abundance [22], and limiting global warming is projected to reduce future dengue incidence. While much attention has been devoted to host mobility to explain spatial transmission patterns [23–27], relatively few studies have explicitly incorporated treatment interventions together with climatic factors. Importantly, temperature shows the strongest correlation with dengue cases among the climate variables [28], including dew point, relative humidity, and

rainfall. Explicitly incorporating treatment is essential to capture the complexity of dengue transmission and guide public health interventions. Dengue in Bangladesh is driven by climate and urban density. Temperatures from 18–30°C favor mosquito growth, and densely populated urban regions show higher infection rates. Transmission peaks when mosquito-friendly conditions last about 10 months [29]. Treatment modeling has also proven important in other diseases, such as rotational fluoroquinolone therapy for ophthalmic MRSA infections [30] and COVID-19 interventions [31,32]. Existing models often rely on generalized global parameters, overlooking local variability in treatment access and effectiveness. Control strategies like insecticide application during peak heat [33] or deployment of Wolbachia-infected mosquitoes [34] have been explored, but typically without integrating treatment dynamics. In Bangladesh, division-level dengue models, separated by care intensity to reflect regional healthcare capability, could represent treatment as a time-varying control that enhances human recovery. The monsoon season (July–August), when rainfall and warmth significantly raise Aedes mosquito populations, should be the focus of temperature impacts. Spatial reality and policy relevance are enhanced by taking into account the recent shift in the dengue burden out of Dhaka [35].

Existing modeling efforts in Bangladesh [32,35–45] and related work [17–21], have largely focused on national-scale dynamics or broad seasonal trends. Despite substantial national-level efforts, division-level heterogeneity in dengue cases, along with climate data and treatment interventions, remains unexplored, a gap our work addresses to support more effective, targeted intervention strategies in Bangladesh's diverse epidemiological landscape. Moreover, similar challenges remain internationally, as studies in Malaysia [46], Brazil [47], Indonesia [48], and elsewhere have developed models combining transmission dynamics and seasonality [13,22,49]. Nevertheless, these models often neglect to guide treatment-focused strategies in densely populated cities like Dhaka. Moreover, most dengue models assume homogeneous recovery, whereas our SITRS framework explicitly distinguishes natural recovery from supportive treatment, thereby capturing differences in healthcare access and effectiveness across provinces in Bangladesh. This separation enables more realistic assessment of treatment-sensitive dynamics, particularly in resource-limited settings. We improved model realism by calibrating our temperature-dependent SITRS framework fitted to Bangladesh dengue surveillance, enabling region-specific forecasting.

Lyapunov-based methods have been used to study the stability of SIR, SEIR [50], and SIRS [48] models; however, their application in more complex frameworks, such as our SITRS model incorporating treatment and re-susceptibility, is limited. This gap enables the use of Lyapunov functions to derive conditions for disease persistence, elimination, and endemicity, as well as to compute the basic reproduction number via the next-generation matrix. Furthermore, calibration of models for Dhaka Metropolitan and the eight provinces (divisions): Barishal, Chittagong, Dhaka, Khulna, Rajshahi, Rangpur, Mymensingh, and Sylhet, remains limited, constraining short-term forecasting essential for timely public health responses. Our SITRS model is calibrated with 2024 dengue data from Bangladesh's Directorate General of Health Services (DGHS) [15] and temperature data [51]. Short-term (7-week) forecasts from July 12 to August 23, 2025, based on data from April 13, 2024, to July 5, 2025, demonstrate that treatment efficacy and mosquito biting rates strongly influence epidemic dynamics. Although dengue lacks a specific antiviral cure, early detection and supportive care—including hydration, symptom management, hospitalization, monitoring, and, when necessary, blood transfusions—are critical for improving outcomes [52]. Variability in healthcare access between urban centers like Dhaka and rural areas is captured by separating natural recovery from recovery due to supportive care, reflecting differences in treatment availability and effectiveness. Our simulations show that improved supportive care substantially reduces infection peaks, highlighting treatment's central role in dengue control in Bangladesh. Mathematical models identify key factors in disease spread. Combining clinical treatment with preventive measures is more effective than single strategies, as improving recovery rates reduces outbreak severity [53]. Stochastic models capture randomness in transmission, especially when environmental reservoirs sustain disease. Combining sanitation with clinical protection offers a more robust risk-mitigation approach [54].

This remaining of the paper is organized as follows. In Section 2, we introduce the dengue transmission model adapted for Bangladesh, together with the supporting data sources and analytical framework. Section 3 presents numerical

simulations assessing the impact of treatment. Section [4] describes data fitting, model calibration, temperature effects, regional variability, and parameter estimation, with a focus on specific applications. Finally, Section [5] provides concluding remarks.

## 2. Model formulation and data sources

We develop a simple deterministic SITRS model to investigate dengue transmission in Bangladesh, incorporating the effects of treatment and temperature. The model uses differential equations to represent human and mosquito interactions, where treatment enhances recovery and temperature influences mosquito behaviour and virus transmission. It helps to assess the impact of climate and healthcare interventions on dengue dynamics in Bangladesh.

### 2.1. Data

The dengue time series datasets, compiled from DGHS epidemiological records [15], comprises daily reported cases at the division level across eight administrative divisions of Bangladesh: Barishal, Chittagong, Dhaka, Khulna, Rajshahi, Rangpur, Mymensingh, and Sylhet, covering the period 2024–2025. To align with standard epidemiological surveillance practices and facilitate model implementation, these daily division-level case counts were aggregated into epidemiological weeks, producing weekly dengue incidence time series for each division. Climate time-series data from reference [51] included variables such as temperature and were available at the district level for all 64 administrative districts. We aggregated these daily climate data from districts to divisions and then averaged them by epidemiological week to ensure both temporal and spatial alignment with the dengue case data. We illustrated this aggregation process in Fig 1, where Fig 1A displays the boundaries of the districts and divisions, while Fig 1B shows the total reported dengue cases per division. Administrative boundary shapefiles were obtained from the Humanitarian Data Exchange (HDX; https://data.humdata.org), providing adm1 (8 divisions) and adm2 (64 districts or Zilas) boundaries for Bangladesh. All maps were created by the authors using Python, primarily with the GeoPandas and Matplotlib libraries.

Fig 1A shows district-level administrative boundaries (left) and aggregated divisions (right) used for processing climate data (e.g., temperature) and dengue case data, with the northern region highlighted for clarity. Fig 1B presents the total number of reported dengue cases per division for 2024 and 2025. Here, we aggregated district-level data to division-level

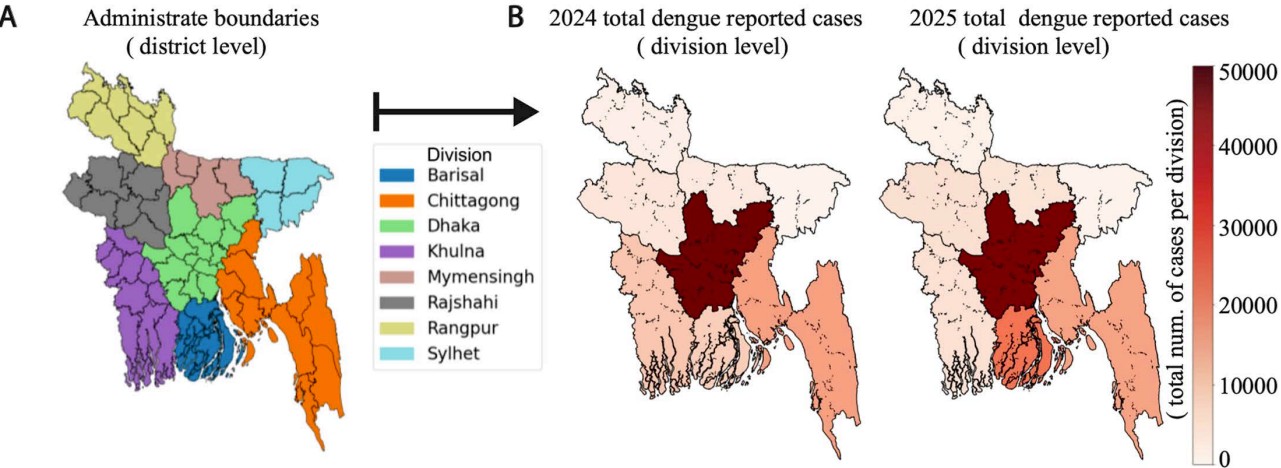

**Fig 1. District and division-level aggregation of climate and dengue case data in Bangladesh.**

data by epidemiological week. The color gradient indicates case burden intensity, from low (light colors) to high (dark colors), as shown by the scale bar.

In Fig 2A displays the time series of dengue cases that were reported in the Dhaka Metropolitan Area from January 1, 2022, to August 23, 2025. The lower-left part of Fig 2B displays a zoomed-in image of the model fit to reported dengue cases. The mosquito bite rate function $b(T)$ is defined using the average temperature ($°C$) over the preceding 14 days. The right part of panel B provides a further zoomed view of the datasets divided into two phases: the fitting phase from April 13, 2024, to July 05, 2025 (EW 15, 2024 to EWS 27, 2025), and the forecasting phase from July 12, 2025, to August 23, 2025 (EW 28, 2025 to EW 34, 2025).

## 2.2. Analyses and descriptions of the model

We explain and assess the following proposed model to gain a better understanding of the dynamics of dengue transmission, for example, Fig 2A, which requires two interacting populations of individuals (the host) and mosquitoes (the vector).

Fig 3 depicts the compartmental model for mosquito-borne disease transmission, illustrating the flow of individuals among the human compartments: susceptible ($S_h$), infected ($I_h$), treated ($T_h$), and recovered ($R_h$) human compartments, as well as the mosquito compartments: susceptible ($S_v$) and infected ($I_v$). Transitions between compartments are governed by infection, recovery, treatment, immunity loss, and demographic processes.

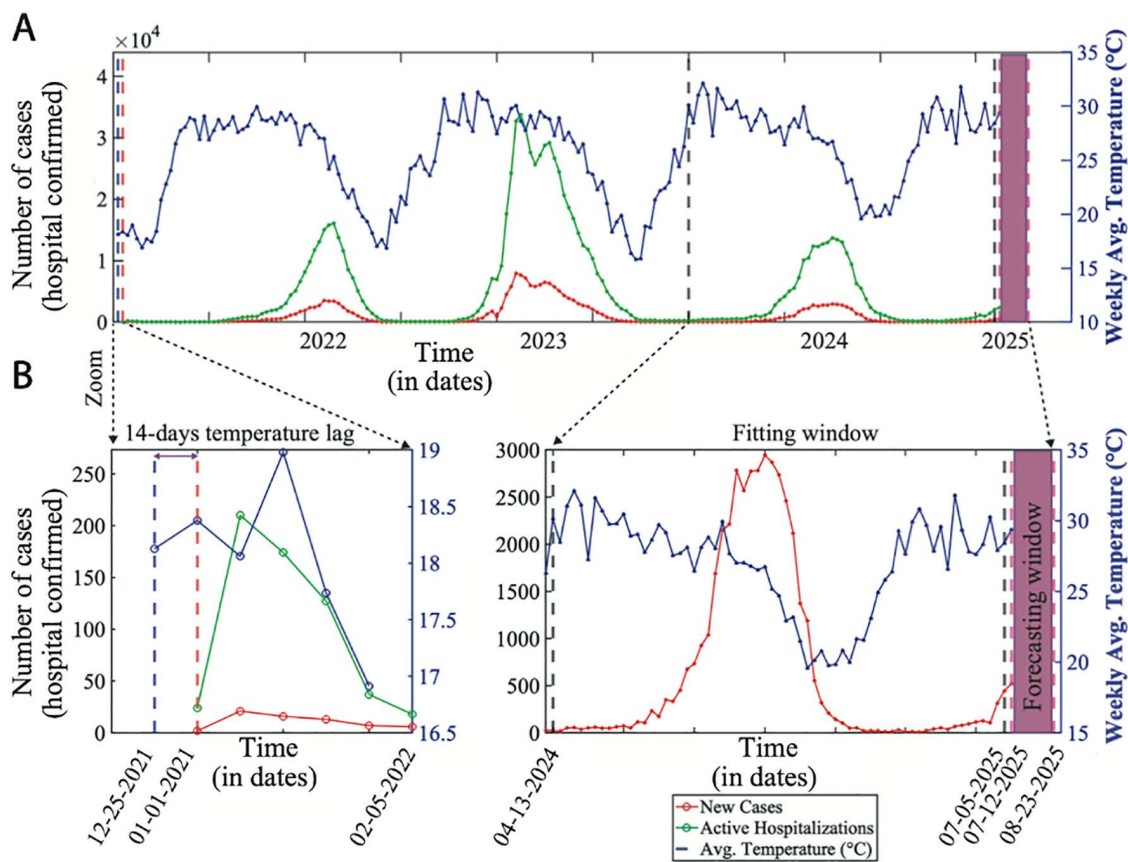

**Fig 2. The Dhaka metropolitan area's weekly dengue incidences (hospital confirmed) and average temperature from 2021 to 2025.**

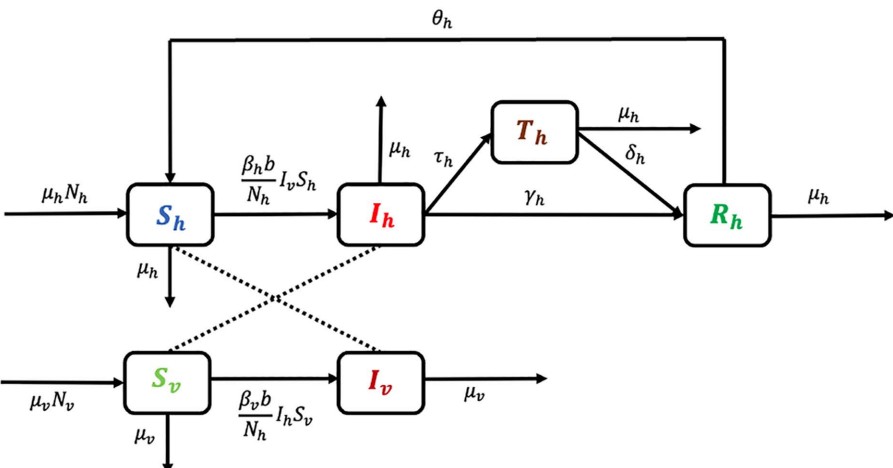

**Fig 3. SITRS model diagram of human and vector population.**

## 2.3. A model of a vector-borne disease with treatment

In this section, we use a simple deterministic approach to model dengue transmission; however, our framework includes treatment as a separate epidemiological compartment (see Fig 3) and distinguishes natural recovery from supportive care, reflecting healthcare disparities between urban and rural regions. For the human compartment, we created a deterministic mathematical model Susceptible-Infected-Treatment-Recovered-Susceptible (SITRS), and other one is, for the mosquito compartment, Susceptible-Infected (SI). The model subdivides the total human population, denoted by $N_h(t)$, into the following sub-classes of individuals who are susceptible to infection with dengue $S_h(t)$, those treated to dengue parasite $T_h(t)$ with dengue symptoms (new infection controlled by mosquito net, quarantine zones and so on), infected individuals $I_h(t)$, those are already carried the virus and are contagious, along with individuals who have recovered $R_h(t)$ with time $t$. The initial conditions used for each compartment are $S_h(0) = S_h^0$, $I_h(0) = I_h^0$, $T_h(0) = T_h^0$, $R_h(0) = R_h^0$ when $t = 0$. Thus, the total human population at time t is given by,

$$N_h(t) = S_h(t) + I_h(t) + T_h(t) + R_h(t).$$

This is considered to remain constant, and people interact at random. Susceptible mosquitoes $S_v(t)$ and infectious mosquitoes $I_v(t)$ comprise the two subpopulations of the entire mosquito population, $N_v(t)$ with time $t$. The mosquitoes have no class recovery and are always contagious. The initial conditions used for each compartment are $S_v(0) = S_v^0$, $I_v(0) = I_v^0$ when $t = 0$. Consequently, at each specific moment, the total size of the mosquito population is given by,

$$N_v(t) = S_v(t) + I_v(t),$$

Which has been determined to remain constant, for individuals integrating at random. The model has been defined by the following nonlinear ordinary equations, which eventually yield the transmission of mosquito and human populations:

$$\frac{dS_h}{dt} = \mu_h N_h - \frac{\beta_h b}{N_h} I_v S_h - \mu_h S_h + \theta_h R_h,$$

$$\frac{dI_h}{dt} = \frac{\beta_h b}{N_h} I_v S_h - (\mu_h + \gamma_h + \tau_h) I_h,$$

$$\frac{dT_h}{dt} = \tau_h I_h - (\delta_h + \mu_h) T_h,$$

(1)

$$\frac{dR_h}{dt} = \gamma_h I_h - (\mu_h + \theta_h) R_h + \delta_h T_h,$$

$$\frac{dS_v}{dt} = \mu_v N_v - \frac{\beta_v b}{N_h} I_h S_v - \mu_v S_v,$$

$$\frac{dI_v}{dt} = \frac{\beta_v b}{N_h} I_h S_v - \mu_v I_v,$$

With conditions $S_h + I_h + T_h + R_h \le N_h$ and $S_v + I_v \le N_v = \frac{A}{\mu_v}$.

The term $\frac{\beta_h b}{N_h} I_v S_h$ represents the rate of new human infections caused by bites from infectious mosquitoes. Dimensionally, the force of infection per susceptible human is $\frac{\beta_h b}{N_h} I_v$, which has units of $time^{-1}$. Multiplying it by $S_h$ yields humans per unit time, consistent with the left-hand side of the equation $\frac{dS_h}{dt}$. In other words, this term quantifies how quickly susceptible humans become infected through contact with infectious mosquitoes, incorporating the number of infectious mosquitoes, the biting frequency, and the likelihood of transmission per bite.

Table 1 summarizes the key variables and parameters used in modeling mosquito-borne disease transmission, including population size, transmission rates, and recovery rates, which together form the basis of the mathematical model.

## 2.4. Equilibrium points

There are two equilibrium points in the SITRS model prescribed in system (1). One is the disease-free equilibrium (DFE) value of $(S_h^0, I_h^0, T_h^0, R_h^0, S_v^0, I_v^0) = E^0 = (N_h, 0, 0, 0, N_v, 0)$, where $N_v = \frac{A}{\mu_v}$ at the disease-free

**Table 1. Variables and parameters.**

| Variables and Parameters | Description |
| --- | --- |
| $N_h$ | Total human population |
| $N_v$ | Total mosquito population |
| $S_h$ | Number of susceptible humans |
| $S_v$ | Number of susceptible mosquitos |
| $I_h$ | Number of infected humans |
| $I_v$ | Number of infected mosquitos |
| $\mu_h$ | Birth and mortality rate of total human population |
| $\mu_v$ | Birth and mortality rate of total mosquito population |
| $b$ | Average biting rate per mosquito per unit time |
| $\gamma_h$ | Natural recovery rate from $I_h$ to $R_h$ |
| $\tau_h$ | Treatment rate from $I_h$ to $T_h$ |
| $\delta_h$ | Recovery rate from treatment |
| $\theta_h$ | The rate of decline in human immunity to disease |
| $\beta_v$ | The chance of a virus transmitting from $I_h$ to $S_v$ |
| $\beta_h$ | The chance of a virus transmitting from $I_v$ to $S_h$ |
| $\beta_h b$ | Interaction the possibility $I_h$ and $S_v$ |
| $\beta_v b$ | Interacting the possibility $I_v$ an $S_h$ |
| $A$ | The mosquito recruitment rate |

equilibrium. Another is $P^{**} = (S_h^{**}, I_h^{**}, T_h^{**}, R_h^{**}, S_v^{**}, I_v^{**}) \in D$ called the endemic equilibrium point, which satisfies $S_h^{**} > 0, I_h^{**} > 0, T_h^{**} > 0, R_h^{**} > 0, S_v^{**} > 0, I_v^{**} > 0$, where

$$S_h^{**} = N_h + I_h^{**} \left( \frac{\theta_h \left[ \gamma_h \left( \delta_h + \mu_h \right) + \tau_h \delta_h \right] - \left( \mu_h + \gamma_h + \tau_h \right) \left( \theta_h + \mu_h \right) \left( \delta_h + \mu_h \right)}{\mu_h \left( \theta_h + \mu_h \right) \left( \delta_h + \mu_h \right)} \right),$$

$$T_h^{**} = \frac{\tau_h I_h^{**}}{\left( \delta_h + \mu_h \right)},$$

$$I_h^{**} = \frac{N_v b^2 \beta_v \beta_h \left[ \theta_h (\gamma_h \left( \delta_h + \mu_h \right) + \tau_h \delta_h) - \left( \mu_h + \gamma_h + \tau_h \right) \left( \theta_h + \mu_h \right) \left( \delta_h + \mu_h \right) \right] - N_h \mu_h \mu_v \left( \theta_h + \mu_h \right) \left( \delta_h + \mu_h \right)}{b \beta_v \left[ (N_h \mu_h \left( \theta_h + \mu_h \right) \left( \delta_h + \mu_h \right) + b N_v \left[ \theta_h (\gamma_h \left( \delta_h + \mu_h \right) + \tau_h \delta_h) - \left( \mu_h + \gamma_h + \tau_h \right) \left( \theta_h + \mu_h \right) \left( \delta_h + \mu_h \right) \right] \right]}$$

$$R_h^{**} = \frac{\gamma_h \left( \delta_h + \mu_h \right) I_h^{**} + \delta_h \tau_h I_h^{**}}{\left( \theta_h + \mu_h \right) \left( \delta_h + \mu_h \right)},$$

$$S_v^{**} = \frac{N_h N_v \mu_v}{(N_h \mu_v + I_h^{**} \beta_v b)},$$

and

$$I_v^{**} = \frac{I_h^{**} b N_v \beta_v}{(N_h \mu_v + I_h^{**} \beta_v b)},$$

Here, the total human population $N_h$ remains constant due to the balance between recruitment and natural mortality.

## 2.5. Basic reproduction number

We can separate the proposed model into two categories disease and non-disease compartments.

(i) Disease compartments

$$\frac{dI_h}{dt} = \frac{\beta_h b}{N_h} I_v S_h - (\mu_h + \gamma_h + \tau_h) I_h,$$

$$\frac{dI_v}{dt} = \frac{\beta_v b}{N_h} I_h S_v - \mu_v I_v,$$

(ii) Non-disease compartments

$$\frac{dS_h}{dt} = \mu_h N_h - \frac{\beta_h b}{N_h} I_v S_h - \mu_h S_h + \theta_h R_h,$$

$$\frac{dT_h}{dt} = \tau_h I_h - \delta_h T_h - \mu_h T_h,$$

$$\frac{dR_h}{dt} = \gamma_h I_h - \left( \mu_h + \theta_h \right) R_h + \delta_h T_h,$$

 

$$\frac{dS_v}{dt} = \mu_v N_v - \frac{\beta_h b}{N_h} I_h S_v - \mu_v s_v,$$

In general, the above disease and non-disease compartments can be written as

$$\frac{\partial x_i}{\partial t} = f_i\left(x_i, y_i\right) - v_i\left(x_i, y_i\right), \quad \frac{\partial y_i}{\partial t} = g\left(x_i, y_i\right),$$

Where $x_i$ and $y_i$ be the subpopulations in disease and non-disease compartments, respectively. Here $f_i$ and $v_i$ are the rate of secondary infections increased in $i-th$ disease compartment and the rate of disease other cases such as treatment, death and recovery decreased in the $i-th$ disease compartment, respectively. Then, we get $f_i = \begin{pmatrix} \frac{\beta_h b}{N_h} I_v S_h \\ \frac{\beta_v b}{N_h} I_h S_v \end{pmatrix}$ and $v_i = \begin{pmatrix} (\mu_h + \gamma_h + \tau_h) I_h \\ \mu_v I_v \end{pmatrix}$. We assume that all new infections are secondary infections and infected by the infected population, and linearizing for the diseases compartment we get,

$x' = (F - V)x$, where $F = \frac{\partial f_i}{\partial x_i}$ and $V = \frac{\partial v_i}{\partial x_i}$ which gives

$$F = \begin{pmatrix} 0 & \frac{\beta_h b}{N_h} S_h \\ \frac{\beta_v b}{N_h} S_v & 0 \end{pmatrix} \text{ and } V = \begin{pmatrix} (\mu_h + \gamma_h + \tau_h) & 0 \\ 0 & \mu_v \end{pmatrix}, \text{ then } V^{-1} = \begin{pmatrix} \frac{1}{\mu_h + \gamma_h + \tau_h} & 0 \\ 0 & \frac{1}{\mu_v} \end{pmatrix}.$$

Following the steps in [55], the next-generation matrix $K = FV^{-1}$ at disease-free equilibrium is as follows:

$$K = FV^{-1} = \begin{pmatrix} 0 & \frac{\beta_h b}{N_h} S_h \\ \frac{\beta_v b}{N_h} S_v & 0 \end{pmatrix} \begin{pmatrix} \frac{1}{\mu_h + \gamma_h + \tau_h} & 0 \\ 0 & \frac{1}{\mu_v} \end{pmatrix} = \begin{pmatrix} 0 & \frac{\beta_v b S_h}{\mu_v N_h} \\ \frac{\beta_h b}{N_h(\mu_h + \gamma_h + \tau_h)} S_v & 0 \end{pmatrix}.$$

Hence, the highest eigenvalue of the next-generation is represents by reproduction number which is given by $\sqrt{\frac{bb\beta_v\beta_h S_h S_v}{\mu_v N_h N_h(\mu_h + \gamma_h + \tau_h)}}$. In the mathematical epidemiology literature, it is customary to denote the reproduction number in the presence of a control strategy, such as treatment, by $R_c^{NG}$ rather than $R_c^{NG}$. Hence, the next-generation basic reproduction number is given by $R_c^{NG} = \sqrt{\frac{bb\beta_v\beta_h S_h S_v}{\mu_v N_h N_h(\mu_h + \gamma_h + \tau_h)}}$. Alternatively, the above expression $R_c^{NG}$, can also be written as

$R_c^{NG} = \sqrt{R_{0h} R_{0v}} = \sqrt{\frac{bb\beta_v\beta_h N_v}{\mu_v N_h(\mu_h + \gamma_h + \tau_h)}}$, where $R_{0h} = \frac{bb\beta_h}{N_h(\mu_h + \gamma_h + \tau_h)}$ and $R_{0v} = \frac{\beta_v N_v}{\mu_v}$ using $S_h = S_h^0 = N_h$ and $S_v = N_v$.

Here $R_{0h}$ represents the average number of infected human individuals by one mosquito that is contagious throughout the likely duration of the illness, and $R_{0v}$, which symbolizes the number of mosquitoes infected by one infectious human during its infectiousness period in a population of totally susceptible mosquitoes. We can also obtain the reproduction number $R_c^{NG}$ via the following Jacobean approach.

## 2.6. Jacobean approach for the computation of $R_c$

In the previous section, $R_c$ gives a threshold condition for the stability of the disease-free equilibrium. In this section, we compute the Jacobean of the system (1) at the disease-free equilibrium value of $\left(S_h^0, I_h^0, T_h^0, R_h^0, S_v^0, I_v^0\right) = E^0 = (N_h, 0, 0, 0, N_v, 0)$, and we present the condition that all eigenvalues of the corresponding characteristic equation must have negative real parts. The Jacobean matrix of the system (1) is defined as follows:

$$J = \begin{bmatrix} -\frac{\beta_h b I_v}{N_h} - \mu_h & 0 & 0 & \theta_h & 0 & -\frac{\beta_h b S_h}{N_h} \\ \frac{\beta_h b I_v}{N_h} & -\mu_h - \gamma_h - \tau_h & 0 & 0 & 0 & \frac{\beta_h b S_h}{N_h} \\ 0 & \tau_h & -\delta_h - \mu_h & 0 & 0 & 0 \\ 0 & \gamma_h & \delta_h & -\mu_h - \theta_h & 0 & 0 \\ 0 & -\frac{\beta_v b S_v}{N_h} & 0 & 0 & -\frac{\beta_v b I_h}{N_h} - \mu_v & 0 \\ 0 & \frac{\beta_v b S_v}{N_h} & 0 & 0 & \frac{\beta_v b I_h}{N_h} & -\mu_v \end{bmatrix}$$

Computing the Jacobean at the disease-free equilibrium point, we obtain

$$J = \begin{bmatrix} -\mu_h & 0 & 0 & \theta_h & 0 & -\beta_h b \\ 0 & -\mu_h - \gamma_h - \tau_h & 0 & 0 & 0 & \beta_h b \\ 0 & \tau_h & -\delta_h - \mu_h & 0 & 0 & 0 \\ 0 & \gamma_h & \delta_h & -\mu_h - \theta_h & 0 & 0 \\ 0 & -\frac{\beta_v b S_v}{N_h} & 0 & 0 & -\mu_v & 0 \\ 0 & \frac{\beta_v b S_v}{N_h} & 0 & 0 & 0 & -\mu_v \end{bmatrix}$$

If we consider $|J - \lambda I| = 0$, we can find the equation

$$\begin{vmatrix} -\mu_h - \lambda & 0 & 0 & \theta_h & 0 & -\beta_h b \\ 0 & -\mu_h - \gamma_h - \tau_h - \lambda & 0 & 0 & 0 & \beta_h b \\ 0 & \tau_h & -\delta_h - \mu_h - \lambda & 0 & 0 & 0 \\ 0 & \gamma_h & \delta_h & -\mu_h - \theta_h - \lambda & 0 & 0 \\ 0 & -\frac{\beta_v b S_v}{N_h} & 0 & 0 & -\mu_v - \lambda & 0 \\ 0 & \frac{\beta_v b S_v}{N_h} & 0 & 0 & 0 & -\mu_v - \lambda \end{vmatrix} = 0,$$

$$(-\mu_h - \lambda)\begin{vmatrix} -\mu_h - \gamma_h - \tau_h - \lambda & 0 & 0 & 0 & \beta_h b \\ \tau_h & -\delta_h - \mu_h - \lambda & 0 & 0 & 0 \\ \gamma_h & \delta_h & -\mu_h - \theta_h - \lambda & 0 & 0 \\ -\frac{\beta_v b S_v}{N_h} & 0 & 0 & -\mu_v - \lambda & 0 \\ \frac{\beta_v b S_v}{N_h} & 0 & 0 & 0 & -\mu_v - \lambda \end{vmatrix} = 0.$$

This implies, $(-\mu_h - \lambda)(-\delta_h - \mu_h - \lambda)(-\mu_h - \theta_h - \lambda)$
$(-\mu_v - \lambda)(\lambda^2 + (\mu_h + \mu_v + \gamma_h + \tau_h)\lambda + (\mu_h + \gamma_h + \tau_h)\mu_v - (\beta_h b)(\beta_v b S_v)) = 0$.

The characteristic polynomial has four obvious eigenvalues are as follows: $\lambda_1 = -\mu_h$, $\lambda_2 = -\delta_h - \mu_h$, $\lambda_3 = -\mu_h - \theta_h$, $\lambda_4 = -\mu_v$. The remaining eigenvalues can be obtained from the following quadratic equation in:

$p(\lambda) = \lambda^2 + a_1 \lambda + a_2$, where the coefficients are the following expression of the parameters:

$a_1 = \mu_h + \mu_v + \gamma_h + \tau_h > 0$ and $a_2 = (\mu_h + \gamma_h + \tau_h)\mu_v - (\beta_h b)(\beta_v b S_v)$. By using the Routh–Hurwitz criterion [56] $a_2 > 0$ guarantees that all the roots of the polynomial $p(\lambda)$ are nonnegative and have negative real parts. This simplifies to $\frac{bb\beta_v\beta_h S_h S_v}{\mu_v N_h N_h(\mu_h + \gamma_h + \tau_h)} < 1$, which is equivalent to the condition $R_c < 1$. Thus, the Routh–Hurwitz criterion then implies that the disease-free equilibrium is locally asymptotically stable if $R_c < 1$.

Here, we observe that the reproduction number obtained via the above Jacobean approach is the square of the reproduction number obtained via the next-generation approach:

$$R_c = \left(R_c^{NG}\right)^2 = \frac{bb\beta_v\beta_h S_h S_v}{\mu_v N_h N_h(\mu_h + \gamma_h + \tau_h)} \approx \frac{bb\beta_v\beta_h S_v}{\mu_v N_h(\mu_h + \gamma_h + \tau_h)} \tag{2}$$

## 2.7. Ethical approval

This study is based on aggregated dengue surveillance data in Bangladesh taken from the Directorate General of Health Services (DGHS), Ministry of Health in Bangladesh (https://old.dghs.gov.bd/index.php/bd/home/5200-daily-dengue-status-report). No confidential information included because analyses were performed at the aggregate level. Therefore, no ethical approval is required.

## 2.8. Further model analysis for the transmission of dengue fever

To ensure the model system is biologically realistic and mathematically valid, we first illustrate the positive invariant of the SITRS model. All the parameters and state variables are assumed to be nonnegative, as shown in system (1). Consequently, the nonnegative octant $R_+^6$ is positively invariant with respect to the system (1). We then analyze the stability conditions to investigate the transmissions dynamics of dengue disease.

**Theorem 1.** Let $(S_h(t) > 0,\; I_h(t) > 0,\; T_h(t) > 0,\; R_h(t) > 0,\; S_v(t) > 0,\; I_v(t) > 0)$ be the solution of the equation system (1) with initial condition $(S_{0h}, I_{0h}, T_{0h}, R_{0h}, S_{0v}, I_{0v})$ on the compact set

$$D = \left\{ \begin{array}{c} (S_h(t), I_h(t), T_h(t), R_h(t), S_v(t), I_v(t)) \epsilon R_+^6, \\ L_1 = S_h + I_h + T_h + R_h \leq N_h, \quad L_2 = S_v + I_v = N_v = \frac{A}{\mu_v} \end{array} \right\} \tag{3}$$

The area $D(Eq\ 3)$ of the model system (1) is a positive invariant that comprises every potential solution of $R_+^6$.

*Proof.* We considered the Lyapunov function [48] of the form in

$$L(t) = (L_1(t), L_2(t)) = (\; S_h + I_h + T_h + R_h,\; S_v + I_v). \tag{4}$$

Equation (4) is satisfied by the derivative of $L(t)$ with respect to time t, which is

$$\frac{dL}{dt} = \left( \frac{dL_1}{dt}, \frac{dL_2}{dt} \right) = \left( \frac{dS_h}{dt} + \frac{dI_h}{dt} + \frac{dT_h}{dt} + \frac{dR_h}{dt}, \frac{dS_v}{dt} + \frac{dI_v}{dt} \right)$$

$$= (\mu_h N_h - \frac{\beta_h b}{N_h} I_v S_h - \mu_h S_h + \theta_h R_h + \frac{\beta_h}{N_h} I_v S_h - (\mu_h + \gamma_h) I_h + \tau_h I_h + \tau_h I_h - \delta_h T_h - \mu_h T_h$$

$$+ \gamma_h I_h - (\mu_h + \theta_h) R_h + \delta_h T_h,\; \mu_v N_v - \frac{\beta_v b}{N_h} I_h S_v - \mu_v S_v + \frac{\beta_v b}{N_h} I_h S_v - \mu_v I_v)$$

$$= (\mu_h N_h - \mu_h (S_h + I_h + T_h + R_h),\; A - \mu(S_v + I_v))$$

$$= (\mu_h N_h - \mu_h L_1,\; A - \mu_v L_2).$$

Thus, we can find

$$\begin{cases} \frac{dL_1}{dt} = \mu_h(N_h - L_1) \leq 0, \text{ for } L_1 \geq N_h, \\ \frac{dL_2}{dt} = A - \mu_v L_2 \leq 0, \text{ for } L_2 \geq \frac{A}{\mu_v}. \end{cases} \tag{5}$$

Therefore, considering [Equation (5)], we were able to determine that $\frac{dL}{dt} \leq 0$, suggesting that $D(Eq\ 3)$ is a positive invariant set. At the same time, the solution to ([Equation 5]) provide $0 \leq (L_1(t), L_2(t)) \leq \left( N_h + L_1(0)e^{-\mu_h t}, \frac{A}{\mu_v} + L_2(0)e^{-\mu_v t} \right)$, where $L_1(0)$ and $L_2(0)$ are the starting condition of $L_1(t)$ and $L_2(t)$ sequentially.

Hence, as $t \to \infty$, $0 \leq (L_1(t), L_2(t)) \leq (N_h, A/\mu_v)$. This confirms that $D(Eq\ 3)$ is a positive invariant set containing all the solutions in $R_+^6$. This proves Theorem 1.

Theorem 1 guarantees the existence of dengue fever transmission in a region in which the dengue fever transmitting virus was formerly absent and then changed when the populations of suspected but not infected $(S_h(t) > 0)$, infected $(I_h(t) > 0)$, treatment $(T_h(t) > 0)$ and recovered $(R_h(t) > 0)$ individuals were found. Additionally, the theorem suggests that

 

further study of the phases of the illness's transmission, according to the SITRS model, can classify an area as either endemic or disease-free.

**Theorem 2.** The disease-free equilibrium $(S_h^0, I_h^0, T_h^0, R_h^0, S_v^0, I_v^0) = E^0 = (N_h, 0, 0, 0, \frac{A}{\mu_v}, 0)$ is globally asymptotically stable in $D(eq.3)$ if $R_c \leq 1$. This may be determined by considering that

$$\mu_h = \frac{\beta_v b}{N_h}, \mu_v = \frac{\beta_h b}{N_h} S_h^0 \tag{6}$$

*Proof.* We considered a Lyapunov function [48] of the form in

$$W(t) = (S_h - S_h^0 \ln S_h) + I_h + T_h + R_h + (S_v - S_v^0 \ln S_v) + I_v \tag{7}$$

For the purpose of to satisfy Equation (7) the derivative of $W(t)$ with respect to time t is

$$\dot{W}(t) = \dot{S}_h \left(1 - \frac{S_h^0}{S_h}\right) + \dot{I}_h + \dot{T}_h + \dot{R}_h + \dot{S}_v \left(1 - \frac{S_v^0}{S_v}\right) + \dot{I}_v$$

$$= \left(\mu_h N_h - \frac{\beta_h b}{N_h} I_v S_h - \mu_h S_h + \theta_h R_h\right)\left(1 - \frac{S_h^0}{S_h}\right) + \frac{\beta_h b}{N_h} I_v S_h - (\mu_h + \gamma_h + \tau_h) I_h + \tau_h I_h - \delta_h T_h - \mu_h T_h$$

$$+ \gamma_h I_h - (\mu_h + \theta_h) R_h + \delta_h T_h + \left(A - \frac{\beta_h b}{N_h} I_h S_v - \mu_v s_v\right)\left(1 - \frac{S_v^0}{S_v}\right) + \frac{\beta_v b}{N_h} I_h S_v - \mu_v I_v$$

$$= \mu_h N_h \left(1 - \frac{S_h^0}{S_h}\right) - \frac{\beta_h b}{N_h} I_v S_h \left(1 - \frac{S_h^0}{S_h}\right) - \mu_h S_h \left(1 - \frac{S_h^0}{S_h}\right) - \theta_h R_h \left(1 - \frac{S_h^0}{S_h}\right)$$

$$+ \frac{\beta_h b}{N_h} I_v S_h - \mu_h I_h - \mu_h T_h - \mu_h R_h - \mu_h R_h + A \left(1 - \frac{S_v^0}{S_v}\right) - \frac{\beta_v b}{N_h} I_h S_v (1 - \frac{S_v^0}{S_v}) - \mu_v s_v \left(1 - \frac{S_v^0}{S_v}\right) + \frac{\beta_v b}{N_h} I_h S_v - \mu_v I_v$$

$$= \mu_h N_h \left(1 - \frac{S_h^0}{S_h}\right) + \frac{\beta_h b}{N_h} I_v S_h \left(\frac{S_h^0}{S_h}\right) + \mu_h S_h^0 \left(\frac{S_h}{S_h^0}\right) - \theta_h R_h \left(\frac{S_h^0}{S_h}\right) - \mu_h I_h - \mu_h T_h - A \left(1 - \frac{S_v^0}{S_v}\right) + \frac{\beta_v b}{N_h} I_h S_v \left(\frac{S_v^0}{S_v}\right)$$

$$+ \mu_v S_v^0 \left(1 - \frac{S_v}{S_v^0}\right) - \mu_v I_v$$

$$= \mu_h N_h \left(1 - \frac{S_h^0}{S_h}\right) + \frac{\beta_h b}{N_h} I_v S_h^0 + \mu_h S_h^0 \left(1 - \frac{S_h}{S_h^0}\right) - \theta_h R_h \left(\frac{S_h^0}{S_h}\right) - \mu_h I_h - \mu_h T_h - \mu_h R_h + A \left(1 - \frac{S_v^0}{S_v}\right) + \frac{\beta_v b}{N_h} I_h S_v^0$$

$$+ \mu_v S_v^0 \left(1 - \frac{S_v}{S_v^0}\right) - \mu_v I_v$$

$$= \mu_h N_h \left(1 - \frac{S_h^0}{S_h}\right) + \mu_h S_h^0 \left(1 - \frac{S_h}{S_h^0}\right) - \theta_h R_h \left(\frac{S_h^0}{S_h}\right) - \mu_h T_h - \mu_h R_h + A \left(1 - \frac{S_v^0}{S_v}\right) + \mu_v S_v^0 \left(1 - \frac{S_v}{S_v^0}\right)$$

$$+ \left(\frac{\beta_v b}{N_h} S_v^0 - \mu_h\right) I_h + \left(\frac{\beta_h b}{N_h} S_h^0 - \mu_v\right) I_v. \tag{8}$$

Considering $S_h^0 = N_h$, $S_v^0 = \frac{A}{\mu_v}$, condition (6) to [Equation (8)] can be expressed as

$$\dot{W}(t) = \mu_h N_h \left( 1 - \frac{S_h^0}{S_h} + 1 - \frac{S_h}{S_h^0} \right) - \theta_h R_h \left( \frac{S_h^0}{S_h} \right) - \mu_h T_h - \mu_h R_h + A \left( 1 - \frac{S_v^0}{S_v} + 1 - \frac{S_v}{S_v^0} \right)$$

$$= -\mu_h N_h \left( 2 - \frac{S_h^0}{S_h} - \frac{S_h}{S_h^0} \right) - \theta_h R_h \left( \frac{S_h^0}{S_h} \right) - \mu_h T_h - \mu_h R_h - A \left( 2 - \frac{S_v^0}{S_v} - \frac{S_v}{S_v^0} \right)$$

$$= -\mu_h N_h \frac{(S_h - S_h^0)^2}{S_h S_h^0} - \theta_h R_h \left( \frac{S_h^0}{S_h} \right) - \mu_h T_h - \mu_h R_h - A \frac{(S_v - S_v^0)^2}{S_v S_v^0}. \tag{9}$$

Equation (9) shows that $\dot{W}(t)$. Using the Lyapunov method [50], the finite sets applicable for the solution are those contained in the largest invariant set, where $S_h = S_h^0$, $R_h = R_h^0 = 0$, and $S_v = S_v^0$, that is, the singleton set $\{S_h^0, I_h^0, T_h^0, R_h^0, S_v^0, I_v^0\}$. This implies that the disease-free equilibrium $S_h^0, I_h^0, T_h^0, R_h^0, S_v^0, I_v^0$ is globally asymptotically stable in $D(Eq\ 3)$. This proves Theorem 2.

This global stability theorem for the disease-free case of the SITRS model explains a stage of the existence of dengue fever, as explained in Theorem 1. Theorem 2 says that if $R_c \leq 1$, then an infected individual will not transmit the infection to others. Thus, dengue fever in this stage can still be controlled and should not be concerned about.

**Theorem 3.** If $R_c > 1$, then the equilibrium status of dengue fever diseases is positively endemic, and equation system (1) exists and is in the global stage asymptotically stable in $D(Eq\ 3)$ by assuming that,

$$S_h^{**} = N_h, \quad S_v^{**} = \frac{A}{\mu_V}, \quad \mu_h = \frac{\beta_v b \left( \mu_h + \gamma_h + \theta_h + \tau_h + \delta_h \right)}{N_h} \frac{1}{r}, \quad \mu_v = \frac{r \beta_h b}{\left( \mu_h + \gamma_h + \theta_h + \tau_h + \delta_h \right)} S_v^{**}, \tag{10}$$

where $r = \frac{\beta_h b}{N_h}$. The mosquito population mortality rate is represented by $\mu_v$, the number $b$ indicates the rate of possibly infectious mosquito bites, $\beta_h b$ expresses the interaction capability between humans and mosquitoes as the vector, and the number of people in the population is likely equivalent to the number of possible cases of dengue disease.

*Proof. We considered the Lyapunov function* [48] *of the form in*

$$V(t) = \left( S_h - S_h^{**} \ln S_h \right) + I_h + T_h + R_h + \frac{\left( \mu_h + \gamma_h + \theta_h + \tau_h + \delta_h \right)}{r S_v^{**}} \left( S_v - S_v^{**} \ln S_v \right) + \frac{\mu_h + \gamma_h + \theta_h + \tau_h + \delta_h}{r S_v^{**}} I_v \tag{11}$$

In order to satisfy [Equation (11)], the derivative of $V(t)$ with respect to time $t$,

$$\dot{V}(t) = \dot{S}_h \left( 1 - \frac{S_h^{**}}{S_h} \right) + \dot{I}_h + \dot{T}_h + \dot{R}_h + \dot{S}_v \left( 1 - \frac{S_v^{**}}{S_v} \right) \left( \frac{\mu_h + \gamma_h + \theta_h + \tau_h + \delta_h}{r S_v^{**}} \right) + \dot{I}_v \frac{\mu_h + \gamma_h + \theta_h + \tau_h + \delta_h}{r S_v^{**}}$$

$$= \mu_h N_h \left( 1 - \frac{S_h^{**}}{S_h} \right) - \frac{\beta_h b}{N_h} I_v S_h \left( 1 - \frac{S_h^{**}}{S_h} \right) - \mu_h S_h \left( 1 - \frac{S_h^{**}}{S_h} \right)$$

$$+ \theta_h R_h \left( 1 - \frac{S_h^{**}}{S_h} \right) + \frac{\beta_h b}{N_h} I_v S_h - \left( \mu_h + \gamma_h + \tau_h \right) I_h + \tau_h I_h - \delta_h T_h - \mu_h T_h$$

$$+ \gamma_h I_h - \left( \mu_h + \theta_h \right) R_h + \delta_h T_h + \left( \frac{\mu_h + \gamma_h + \theta_h + \tau_h + \delta_h}{r S_v^{**}} \right) A \left( 1 - \frac{S_v^{**}}{S_v} \right) - \left( \frac{\mu_h + \gamma_h + \theta_h + \tau_h + \delta_h}{r S_v^{**}} \right)$$

$$\frac{\beta_v b}{N_h} I_h S_v \left( 1 - \frac{S_v^{**}}{S_v} \right) - \left( \frac{\mu_h + \gamma_h + \theta_h + \tau_h + \delta_h}{r S_v^{**}} \right) \mu_v S_v \left( 1 - \frac{S_v^{**}}{S_v} \right) + \left( \frac{\mu_h + \gamma_h + \theta_h + \tau_h + \delta_h}{r S_v^{**}} \right) \frac{\beta_v b}{N_h} I_h S_v$$

$$- \left( \frac{\mu_h + \gamma_h + \theta_h + \tau_h + \delta_h}{r S_v^{**}} \right) \mu_v I_v$$

$$= \mu_h N_h \left(1 - \frac{S_h^{**}}{S_h}\right) + \theta_h R_h - \theta_h R_h \left(\frac{S_h^{**}}{S_h}\right) - \mu_h S_h \left(1 - \frac{S_h^{**}}{S_h}\right) - \frac{\beta_h b}{N_h} I_v S_h - \frac{\beta_h b}{N_h} I_v S_h^{**}$$

$$+ \frac{\beta_h b}{N_h} I_v S_h - \mu_h I_h - \mu_h T_h - \mu_h R_h - \theta_h R_h \ + A\left(\frac{\mu_h + \gamma_h + \theta_h + \tau_h + \delta_h}{r S_v^{**}}\right) - A\left(\frac{\mu_h + \gamma_h + \theta_h + \tau_h + \delta_h}{r S_v}\right)$$

$$- \frac{\beta_v b}{N_h} I_h S_v \left(\frac{\mu_h + \gamma_h + \theta_h + \tau_h + \delta_h}{r S_v^{**}}\right) + \frac{\beta_v b}{N_h} I_h \left(\frac{\mu_h + \gamma_h + \theta_h + \tau_h + \delta_h}{r}\right)$$

$$- \left(\frac{\mu_h + \gamma_h + \theta_h + \tau_h + \delta_h}{r}\right) \frac{\beta_h b}{N_h} I_h - \left(\frac{\mu_h + \gamma_h + \theta_h + \tau_h + \delta_h}{r}\right) \frac{\beta_h b}{N_h} I_v \left(\frac{S_v}{S_v^{**}}\right)$$

$$- \left(\frac{\mu_h + \gamma_h + \theta_h + \tau_h + \delta_h}{r S_v^{**}}\right) \mu_v I_v$$

$$= \mu_h N_h \left(1 - \frac{S_h^{**}}{S_h}\right) + \mu_h S_h^{**} \left(1 - \frac{S_h^{**}}{S_h}\right) - \theta_h R_h \left(\frac{S_h^{**}}{S_h}\right) - \mu_h R_h - \frac{\beta_h b}{N_h} I_v S_h^{**} - \mu_h I_h - \mu_h T_h$$

$$+ A\left(\frac{\mu_h + \gamma_h + \theta_h + \tau_h + \delta_h}{r S_v^{**}}\right) - A\left(\frac{\mu_h + \gamma_h + \theta_h + \tau_h + \delta_h}{r S_v}\right) + \frac{\beta_v b}{N_h} I_h \left(\frac{\mu_h + \gamma_h + \theta_h + \tau_h + \delta_h}{r}\right)$$

$$- \left(\frac{\mu_h + \gamma_h + \theta_h + \tau_h + \delta_h}{r}\right) \left(\frac{S_v}{S_v^{**}}\right) \mu_v + \mu_v \left(\frac{\mu_h + \gamma_h + \theta_h + \tau_h + \delta_h}{r}\right) - \left(\frac{\mu_h + \gamma_h + \theta_h + \tau_h + \delta_h}{r S_v^{**}}\right) \mu_v I_v \tag{12}$$

Substituting Equation (10) into Equation (12), we can find

$$= \mu_h N_h \left(2 - \frac{S_h^{**}}{S_h} - \frac{S_h}{S_h^{**}}\right) - \theta_h R_h \left(\frac{S_h^{**}}{S_h}\right) - \mu_h R_h + \left(\frac{\beta_h b}{N_h} S_h^{**} - \left(\frac{\mu_h + \gamma_h + \theta_h + \tau_h + \delta_h}{r S_v^{**}}\right) \mu_v\right) I_v - \mu_h T_h$$

$$+ \left(\frac{\beta_h b}{N_h} \left(\frac{\mu_h + \gamma_h + \theta_h + \tau_h + \delta_h}{r}\right) - \mu_h\right) I_h + \left(\frac{\mu_h + \gamma_h + \theta_h + \tau_h + \delta_h}{r S_v^{**}}\right) \mu_v S_v^{**}$$

$$- \left(\frac{\mu_h + \gamma_h + \theta_h + \tau_h + \delta_h}{r S_v}\right) \mu_v S_v^{**} + \mu_v \left(\frac{\mu_h + \gamma_h + \theta_h + \tau_h + \delta_h}{r}\right) - \left(\frac{\mu_h + \gamma_h + \theta_h + \tau_h + \delta_h}{r}\right) \left(\frac{S_v}{S_v^{**}}\right) \mu_v$$

$$= \mu_h N_h \left(2 - \frac{S_h^{**}}{S_h} - \frac{S_h}{S_h^{**}}\right) - \theta_h R_h \left(\frac{S_h^{**}}{S_h}\right) - \mu_h R_h - \mu_h T_h + \frac{\mu_v}{r} 2(\mu_h + \gamma_h + \theta_h + \tau_h + \delta_h)$$

$$- \mu_v \left(\frac{\mu_h + \gamma_h + \theta_h + \tau_h + \delta_h}{r}\right) \left(\frac{S_v}{S_v^{**}}\right)$$

$$- - \mu_v \left(\frac{\mu_h + \gamma_h + \theta_h + \tau_h + \delta_h}{r}\right) \left(\frac{S_v^{**}}{S_v}\right) \mu_h N_h \left(2 - \frac{S_h^{**}}{S_h} - \frac{S_h}{S_h^{**}}\right)$$

$$- \mu_v \left(\frac{\mu_h + \gamma_h + \theta_h + \tau_h + \delta_h}{r}\right) \left(2 - \frac{S_h^{**}}{S_h} - \frac{S_h}{S_h^{**}}\right) - \theta_h R_h$$

$$= -\mu_h N_h \left(\frac{(S_h - S_h^{**})^2}{S_h S_h^{**}}\right) - \mu_v \left(\frac{\mu_h + \gamma_h + \theta_h + \tau_h + \delta_h}{r}\right) \left(\frac{(S_h - S_h^{**})^2}{S_h S_h^{**}}\right) - \theta_h R_h \left(\frac{S_h^{**}}{S_h}\right) - \mu_h R_h - \mu_h T_h. \tag{13}$$

Equation (13) shows that $\dot{V}(t) \leq 0$ for all $S_h^{**}, I_h^{**}, T_h^{**}, R_h^{**}, S_v^{**}, I_v^{**} \epsilon D, S_h^{**}, I_h^{**}, T_h^{**}, R_h^{**}, S_v^{**}, I_v^{**}$ and $\dot{V}(t) = 0$ for $S_h = S_h^{**}$, $I_h = I$, $T_h = T_h^{**}$, $R_h = R_h^{**}$, $S_v = S_v^{**}$, and $I_v = I_v^{**}$. Then the endemic equilibrium point $P^{**}$ is a set of positive invariant of system (1) that is contained in

$$L = \left\{ (S_h(t), I_h(t), T_h(t), R_h(t), S_v(t), I_v(t)),\ S_h = S_h^{**},\ I_h = I,\ T_h = T_h^{**}, R_h = R_h^{**}, S_v = S_v^{**},\ I_v = I_v^{**} \right\}$$

Using the asymptotical stability theorem, positive endemic equilibrium $P^{**}$ is globally asymptotically stable in $D(Eq\ 3)$. This proves Theorem 3.

 The global stability theorem for the model SITRS in this stage tells if an individual is infected with dengue fever with $R_c > 1$; then, the individual will likely infect at least another individual. Thus, the dengue fever in this situation has been endemic, uncontrolled, and threatening for the human population within the region.

### 2.9. Model parameters and initial conditions

We assign baseline model parameters to guide numerical simulations, summarized in Table 2. These parameters include population sizes, transmission rates, recovery rates, and mosquito biting frequencies, obtained from the literature or assumed when necessary. A subset of parameters ($v = [\tau_h, \beta_h, \gamma_h, \beta_v, b, \rho]$) were later re-estimated by fitting the model to weekly reported dengue case data from the Dhaka Metropolitan area and the eight administrative divisions of Bangladesh (April 13, 2024 – August 23, 2025) to better reflect observed transmission dynamics. This two-step approach first allows us to assess model disease dynamics using literature-based values, and then to refine the parameters using observed data. Here, we assign baseline model parameters to guide numerical simulations, summarized in Table 2.

 Following Table 2, the initial conditions for simulations were set as follows: the total susceptible human population, $S_h(0)$, was set to the actual population size for each division (S1 Table). The numbers of infected ($I_h(0) = 1$), treated ($T_h(0) = 0$), and recovered ($R_h(0) = 0$) humans were initialized according to model assumptions, with $R_h(0) = 0$. Mosquito compartments were initialized with $S_v(0) = N_v - I_v(0)$ and $I_v(0) = 10$ set according to model assumptions. For calibration,

**Table 2. Presenting the simulation's parameter values.**

| Variable | Value | Unit | $R_c$ | Source | Value | Unit | $R_c$/Week | Source |
|---|---|---|---|---|---|---|---|---|
| $N_h$ | 17295319 | Unitless | 3.0456 (based on column 2) | Varies across locations | 169828921 | Unitless | 160.3525 (based on column 6) | [57] |
| $N_v$ | 19080000 | Unitless | | [58] | 19080000 | Unitless | | [58] |
| $\mu_h$ | 2.9586e-04 | $Week^{-1}$ | | Assumed | 0.0143 | $Year^{-1}$ | | [58] |
| | | | | | 0.000275 | $Week^{-1}$ | | |
| $\mu_v$ | 0.7019 | $Week^{-1}$ | | Assumed | 0.032300 | $Month^{-1}$ | | [48] |
| | | | | | 0.008 | $Week^{-1}$ | | |
| $\gamma_h$ | 0.143 | $Week^{-1}$ | | Varies | 0.328833 | $Month^{-1}$ | | [48] |
| | | | | | 0.0822 | $Week^{-1}$ | | |
| $\tau_h$ | 0.03940 | $Week^{-1}$ | | Varies | 0.53 | $Day^{-1}$ | | [59] |
| | | | | | 3.71 | $Week^{-1}$ | | |
| $\delta_h$ | 0.10000 | $Week^{-1}$ | | Assumed | 0.99977 | $Day^{-1}$ | | [60] |
| | | | | | 6.99839 | $Week^{-1}$ | | |
| $\theta_h$ | 0.00274 | $Week^{-1}$ | | Assumed | 0.575000 | $Month^{-1}$ | | [48] |
| | | | | | 0.14375 | $Week^{-1}$ | | |
| $\beta_v$ | 0.85 | Unitless | | Varies | 0.375 | Unitless | | [61] |
| $\beta_h$ | 0.85 | Unitless | | Varies | 0.375 | Unitless | | [61] |
| b | 0.70 | $Week^{-1}$ | | Varies | 0.8 | $Day^{-1}$ | | [61] |
| | | | | | 5.6 | $Week^{-1}$ | | |
| $\rho$ | 0.1 | Unitless | | Varies | 0.1 | Unitless | | [62] |

$I_h(0)$ was set based on reported cases in the first epidemiological week, with $\rho = 1/10$. In divisions with zero reported cases (Rajshahi and Rangpur), a single infectious individual was assumed. These specifications ensure reproducibility of both simulations and calibration.

Using the baseline parameters in Table 2, the invasion threshold $R_c$ is 3.0456 for Bangladesh, which is comparable to literature estimates from other countries (e.g., 6.6 for Brazil, 1–9 for Colombia, $1.292 - 1.753$ (*average* 1.54) for Ahmedabad (India), 26.48 for Indonesia, $\approx 1.01$–1.02 for Singapore, and 2.396 for Cape Verde; S4 Table. Here, $R_c$ represents the basic reproduction threshold that determines whether dengue transmission can invade the population. Literature-based parameters produce a higher $R_c$ of 160.3525. Thus, using the baseline parameters is reasonable for numerical simulations. To adjust for limitations in the assumed values, we also applied a least-squares fitting approach to re-estimate select parameters, as detailed in S4 Table.

While many parameters influence dengue transmission, this study focuses on those that directly affect the basic reproduction number $R_c$. By conducting a sensitivity analysis with respect to $R_c$, we ensure that the parameters examined are most relevant to understanding how changes in biological and epidemiological factors affect dengue spread.

## 2.10. Sensitivity index and Partial Rank Correlation Coefficient (PRCC) analysis

Sensitivity analysis in a mathematical model measure how the model's output responds to changes in its input parameters. It provides insight into the model's robustness and helps identify which parameters most significantly influence the output. Various methods can be used for sensitivity analysis, all aimed at quantifying the relationship between input parameters and model outcomes. In this study, we applied both the normalized forward sensitivity index and **Partial Rank Correlation Coefficient (PRCC)** to evaluate the influence of model parameters on the control reproduction number $R_c$ and model dynamics.

**Normalized Forward Sensitivity Index.** The normalized forward sensitivity index [17] of $R_c$ with respect to a parameter $q_i$ is defined as:

$$S_{q_i}^{R_c} = \frac{\partial R_c}{\partial q_i} \times \frac{q_i}{R_c},$$

where, $q_i$ represents the parameters that are involved in the expression of the basic reproduction number $R_c$. The sensitivity index quantifies the relative change in $R_c$ resulting from a relative change in $q_i$. For example, a sensitivity of 1 for $R_c$ with respect to $\beta_h$ means that a 100% increase in $\beta_h$ would lead to a 100% increase in $R_c$, highlighting the direct influence of that parameter on disease transmission. The computed sensitivity indices for the model parameters are:

$$\frac{\partial R_c}{\partial \beta_h} \times \frac{\beta_h}{R_c} = 1, \quad \frac{\partial R_c}{\partial \mu_h} \times \frac{\mu_h}{R_c} = -\frac{\mu_h}{\mu_h + \gamma_h + \tau_h}, \quad \frac{\partial R_c}{\partial \gamma_h} \times \frac{\gamma_h}{R_c} = -\frac{\gamma_h}{\mu_h + \gamma_h + \tau_h}, \quad \frac{\partial R_c}{\partial b} \times \frac{b}{R_c} = 2,$$

$$\frac{\partial R_c}{\partial \tau_h} \times \frac{\tau_h}{R_c} = -\frac{\tau_h}{\mu_h + \gamma_h + \tau_h}, \quad \frac{\partial R_c}{\partial \beta_v} \times \frac{\beta_v}{R_c} = 1, \quad \frac{\partial R_c}{\partial \mu_v} \times \frac{\mu_v}{R_c} = -1,$$

And, $\beta_h = +1$, $\mu_h = -0.000162$, $\gamma_h = -0.782$, $\tau_h = -0.282$, $b = +2$, $\beta_v = +1$, $\mu_v = -1$.

In practical terms, positive sensitivity values indicate that increasing the parameter will increase $R_c$, while negative values indicate a decreasing effect. Among the parameters, $\beta_h$ and the biting rate $b$ have the largest positive influence on $R_c$, whereas the natural recovery rate $\gamma_h$ and vector mortality rate $\mu_v$ negatively affect $R_c$. The human mortality rate $\mu_h$ has a negligible influence.

**Partial Rank Correlation Coefficient (PRCC).** In addition to the normalized sensitivity index, we computed the PRCC values of the model parameters ($\beta_h, \beta_v, b, \gamma_h, \tau_h, \mu_h, \mu_v$) to assess their impact on model outputs while accounting for

nonlinear relationships and interactions among parameters. PRCC values range from −1 to +1, where a positive value indicates a parameter is positively associated with the model output, and a negative value indicates an inverse relationship.

**Interpretation.** By combining the normalized sensitivity index and PRCC analysis, we can clearly identify which parameters most affect disease dynamics and the control reproduction number. For example, a sensitivity index of +1 for $\beta_h$ indicates that interventions reducing human-to-human transmission will proportionally reduce $R_c$. Similarly, the PRCC analysis highlights the same parameters as key drivers of model output, providing robust guidance for targeting control measures and prioritizing parameter estimation efforts.

The **left panel** in Fig 4 shows the normalized sensitivity indices for the model parameters. Positive values indicate parameters that increase $R_c$ when they increase, while negative values indicate parameters that decrease $R_c$. For the human class, the biting rate $b$ and transmission rate $\beta_h$ have the largest positive impact, whereas the natural recovery rate $\gamma_h$ has a negative effect. For the vector class, the transmission rate $\beta_v$ positively influences outcomes, while the mosquito birth and mortality rate $\mu_v$ negatively affects them. For example, a 100% increase in $\beta_h$ would result in an approximately 100% increase in $R_c$. The **right panel** shows the PRCC values of model parameters with respect to model output. Positive PRCC values indicate a favourable association, while negative values indicate an adverse effect. Parameters $b, \beta_h$, and $\beta_v$ are strongly positively correlated with the output, whereas $\gamma_h$ and $\mu_v$ are strongly negatively correlated. Parameters $\mu_h$ and $\tau_h$ show minimal or moderate influence. Overall, both sensitivity index and PRCC analyses consistently indicate that $b, \beta_h, \beta_v, \gamma_h$, and $\mu_v$ are the most influential parameters, while $\mu_h$ has comparatively little impact on model outcomes.

## 3. Numerical simulations

In this section, we conduct numerical simulation to analyse the disease-related mortality, treatment, loss of immunity, recovery rate, and human and vector population impact on dengue prevalence. Our analysis focuses on the most sensitive parameters (see Fig 4), as they have the greatest impact on disease dynamics and provide the most meaningful improvements when fitted to data. In order to strengthen the analytical results, the **MATLAB** programming language is used. The stability results for the model equilibrium were obtained by using different initial conditions for each population. The extinction of dengue disease is anticipated in the case when the basic rate of reproduction $R_c$ is smaller than one. When dengue disease is still present in the human population, it is because that basic reproduction number $R_c$ is greater than one (see Fig 5).

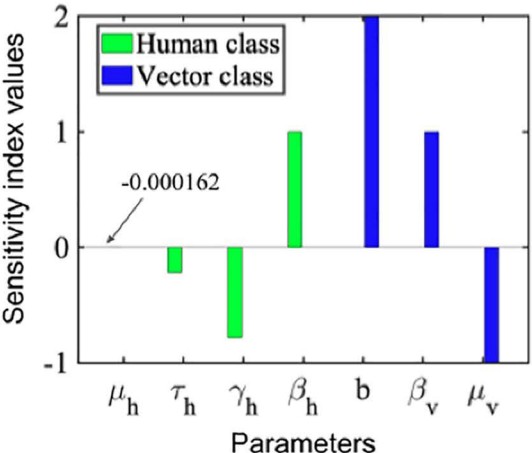 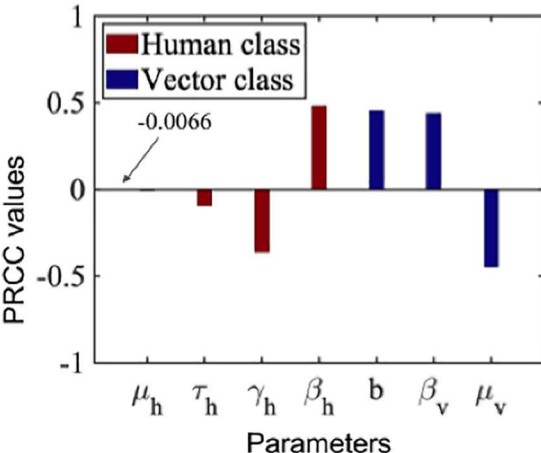

**Fig 4. Sensitivity index (left) and Partial Rank Correlation Coefficient (right) for model parameters in human and vector classes, illustrating the relative influence of each parameter on model outcomes.**

Fig 5A, using the baseline parameter values listed in Table 2, except that the human population is taken as ($N_h$ = 62700000), the basic reproduction number is ($\boldsymbol{R_c}$= **0.8401 < 1**). This indicates that the disease-free equilibrium is locally asymptotically stable and the infection eventually dies out over time. Fig 5B, for the modified parameter set ($\mu_h$ = 0.01, $\gamma_h$ = 0.05, $\tau_h$ = 0.07, $\delta_h$ = 0.05, $\mu_v$ = 0.02, $\theta_h$ = 0.01, $N_h$ = 15295319, $N_v$ = 20080000, $b$ = 0.07, $\beta_h$ = 0.755, $\beta_v$ = 0.855), the reproduction number increases to ($\boldsymbol{R_c}$= **1.277 > 1**). This implies that the disease-free equilibrium becomes unstable and the system converges to an endemic equilibrium where the infection persists in the population.

### 3.1. Disease dynamics under different scenarios

We conducted numerical simulations to analyze disease dynamics over time, which are governed by key factors such as transmission rates, recovery rates, population structure, and interventions, such as treatment. A central concept in understanding disease behavior is the basic reproduction number, which indicates the expected number of secondary infections generated by a single infectious individual in a fully susceptible population. As illustrated in Fig 5, when the effective reproduction number falls below one ($\boldsymbol{R_c}$ < 1), the disease dies out, whereas for $\boldsymbol{R_c}$ > 1, the system reaches an endemic equilibrium. The simulations also reveal distinct patterns in disease treatment under varying conditions.

Fig 6 illustrates the colour map of the theoretical basic reproduction number $R_c^{NG}$ as a function of $\beta_h$ and $\beta_v$, both ranging from 0 to 1, along with plots of the theoretical threshold at $R_0$ = 1. The other parameters are fixed as follows: $\mu_h$ = 2.9586$e$ − 04, $\mu_v$ = 0.7019, $\tau_h$ = 0.03940, $\gamma_h$ = 0.143, $b$ = 0.70 and population sizes $N_h$ = 17295319, $N_v$ = 19080000, are fixed. The simulation highlights the disease dynamics in humans (red solid line) and vectors (blue solid line), demonstrating the potential for both short-term and long-term outbreaks. Most importantly, the duration and intensity of outbreaks depend on the values of $\beta_h$ and $\beta_v$ within the simulation. The simulated parameters and initial values are same to those listed in Table 2. Fig 6A, Fig 6C and Fig 6D show the temporal evolution of infected humans (red curves) and vectors (blue curves) under different transmission parameters $\beta_h$ (human) and $\beta_v$ (vector). Fig 6B depicts the case dynamics when transmission rates produce epidemic peaks. Fig 6E presents a heatmap of the basic reproduction number $R_c^{NG}$ as a function of $\beta_h$ (human) and $\beta_v$ (vector), with the theoretical threshold $R_c^{NG}$ = 1 indicated by the dashed line. Specific parameter combinations are annotated on the heatmap, illustrating their relative positions in the transmission landscape.

### 3.2. Epidemic and non-epidemic parametric regions

To understand the parameter space and identify conditions under which outbreaks are likely to occur, we analyzed the effects of key epidemiological parameters. Epidemic regions are characterized by parameter combinations that sustain transmission, whereas non-epidemic regions correspond to parameter sets in which infections die out or remain at low levels. The results indicate that parameters such as the treatment rate, transmission rates between hosts and vectors, and vector biting rate strongly influence both the likelihood and magnitude of outbreaks. Interventions targeting these factors can shift the system from an epidemic to a non-epidemic regime, effectively reducing the size and duration of potential

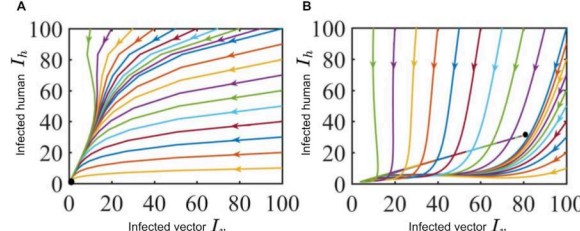

**Fig 5. Impact of disease dynamics for two different parameter systems for 10 years.**

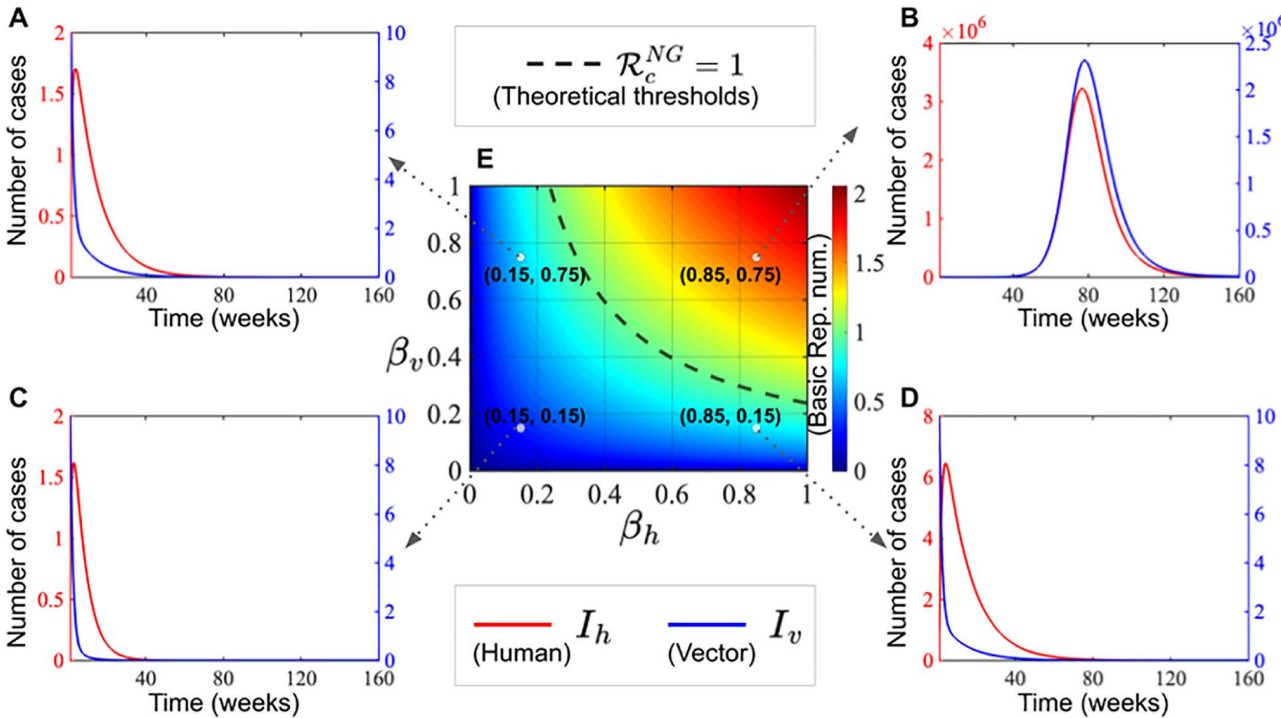

**Fig 6. Transmission dynamics of dengue modeled through human and vector infections over time (weeks) and parameter space analysis.**

outbreaks. To incorporate these numerical simulations, an outbreak is defined, following [62–64], as occurring when the maximum value of the function $I(t)$ within a specified time interval $\tau_h$ exceeds the initial infected value by at least one infected individual. This parametric analysis provides a comprehensive view of the thresholds separating epidemic from non-epidemic behavior, highlighting critical intervention points and the potential impact of targeted control strategies on disease dynamics (see Fig 6).

Fig 7 illustrates the epidemic region in yellow, while the non-epidemic parametric region is shown in blue in Fig 7A, Fig 7B, Fig 7C and Fig 7D. The epidemic curves in Fig 7E highlight human disease dynamics (red solid line) and vectors (blue solid line), demonstrating the potential outbreaks. Most importantly, the results demonstrate that $\tau_h, \beta_h, \beta_v$ and $b$ can reduce epidemic outbreaks. Simulations were conducted over the time domain [0, 160] with a step size of 0.02, using parameter values listed in Table 2.

### 3.3. Impact of treatment on infection dynamics across different risk scenarios

To explore the impact of treatment on infection dynamics in humans and mosquitoes, we systematically varied the treatment rate ($\tau_h$) and the recovery rate from treatment ($\delta_h$) across a range of 0–1 (see Fig 7). The natural recovery rate ($\gamma_h$) was fixed to reflect typical recovery without treatment, allowing us to focus on how supportive care affects outcomes. This approach enables us to evaluate how varying levels (low, low-moderate, moderate, moderate-high, high, and very high infection scenarios) of treatment coverage and effectiveness impact the spread of the disease (see Fig 8).

Here, we defined six distinct scenarios representing varying levels of infection risk. Each scenario differs in terms of the mosquito biting rate and the transmission rates between humans and mosquitoes, while all start with an initial infected mosquito population of ten. Specifically, $\beta_h$ is the chance of a virus transmitting from an infected mosquito ($I_v$) to a susceptible human ($S_h$), and $\beta_v$ is the chance of a virus transmitting from an infected human ($I_h$) to a susceptible mosquito ($S_v$).

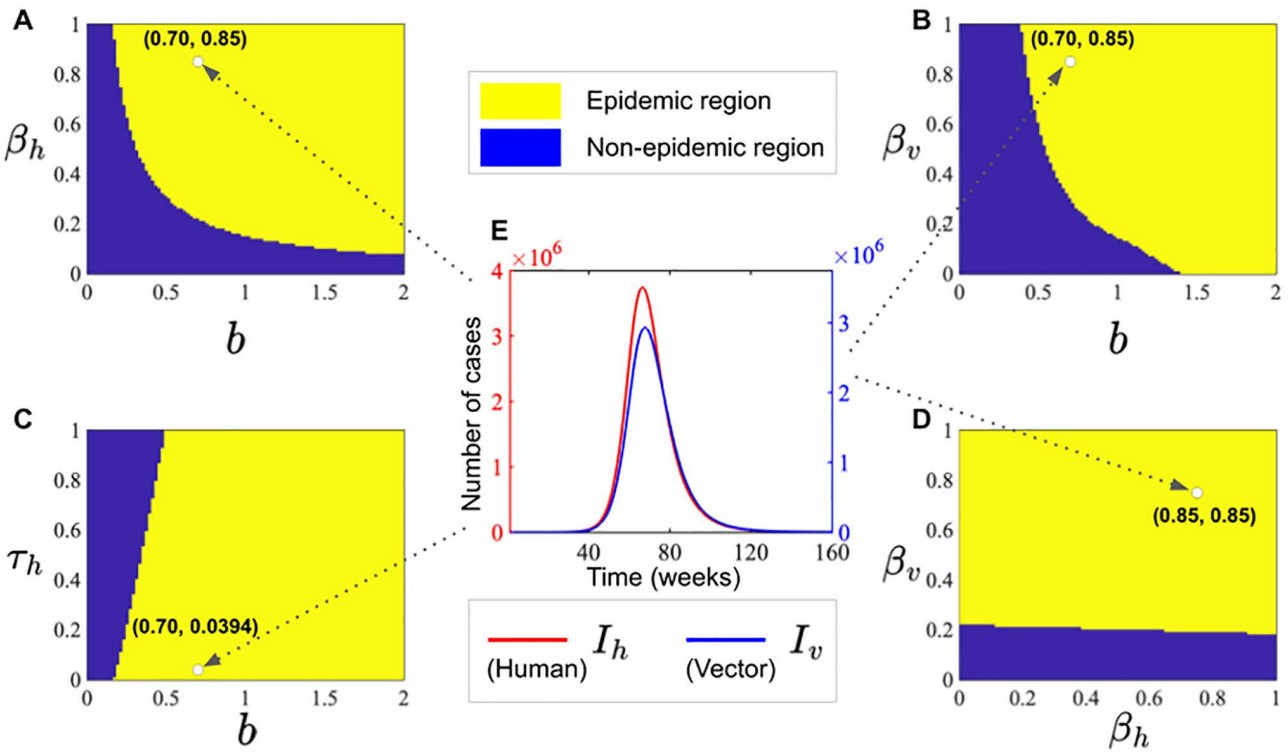

**Fig 7. Epidemic vs. non-epidemic parametric regions and disease dynamics in epidemic regions (Panel E) and non-epidemic regions (Panel A-D) for humans and vectors.**

This enables us to investigate how varying levels of treatment coverage and effectiveness impact the spread of the disease. We defined six infection scenarios with increasing transmission intensity. The Low Infection scenario uses $b$ = 0.7, $\beta_h$ = 0.3 and $\beta_v$ = 0.15. The Low-Moderate scenario increases these values to $b$ = 0.85, $\beta_h$ = 0.5 and $\beta_v$ = 0.2. The Moderate scenario uses $b$ = 1.0, $\beta_h$ = 0.6 and $\beta_v$ = 0.3. In the Moderate-High scenario, the parameters are $b$ = 1.25, $\beta_h$ = 0.7 and $\beta_v$ = 0.4. The High scenario is defined by $b$ = 1.5, $\beta_h$ = 0.85 and $\beta_v$ = 0.5. Finally, the Very High scenario features the most intense transmission dynamics, with $b$ = 1.7, $\beta_h$ = 1.0 and $\beta_v$ = 0.6.

While the numeric values were assumed, they are epidemiologically plausible. In particular, in the Moderate-High and Very High scenarios, the biting rate exceeds 1. This is both mathematically valid and meaningful because $b$ is a rate parameter, not a probability. A value of $b$ = 1 corresponds to each mosquito biting once per day, while $b$ = 1.25 indicates that, on average, each mosquito bites 1.25 times per week. Higher biting rates increase contact between infected mosquitoes and susceptible humans, amplifying dengue transmission and contributing to elevated reproduction numbers. Such increases can arise from favourable environmental conditions, high vector density, or increased host availability [65].

Fig 8 shows a multidimensional sensitivity plot of the parameters $\tau_h$ and $\delta_h$ across the six levels of infection severity. In Fig 8A, there is a strong tilt to the parameter $\tau_h$, where the changes to the parameter $\delta_h$ don't really do much to the outcome. As the parameter $\tau_h$ approaches zero, the peak outcome goes into the red, and then quickly drops back to the baseline as $\tau_h$ increases. The level of infection severity is what really makes the outcome spike, going from 2.5 to over $10 * 10^6$ from the level labeled "Low" to the level labeled "Very High." Fig 8B is a bit different for the vectors, where in the early stages of the infection, represented by the levels labeled "Low" and "Low-Moderate," the parameters don't seem to make much difference, where the heatmap is flat, but after the level labeled "Moderate," the sensitivity to the parameter $\tau_h$

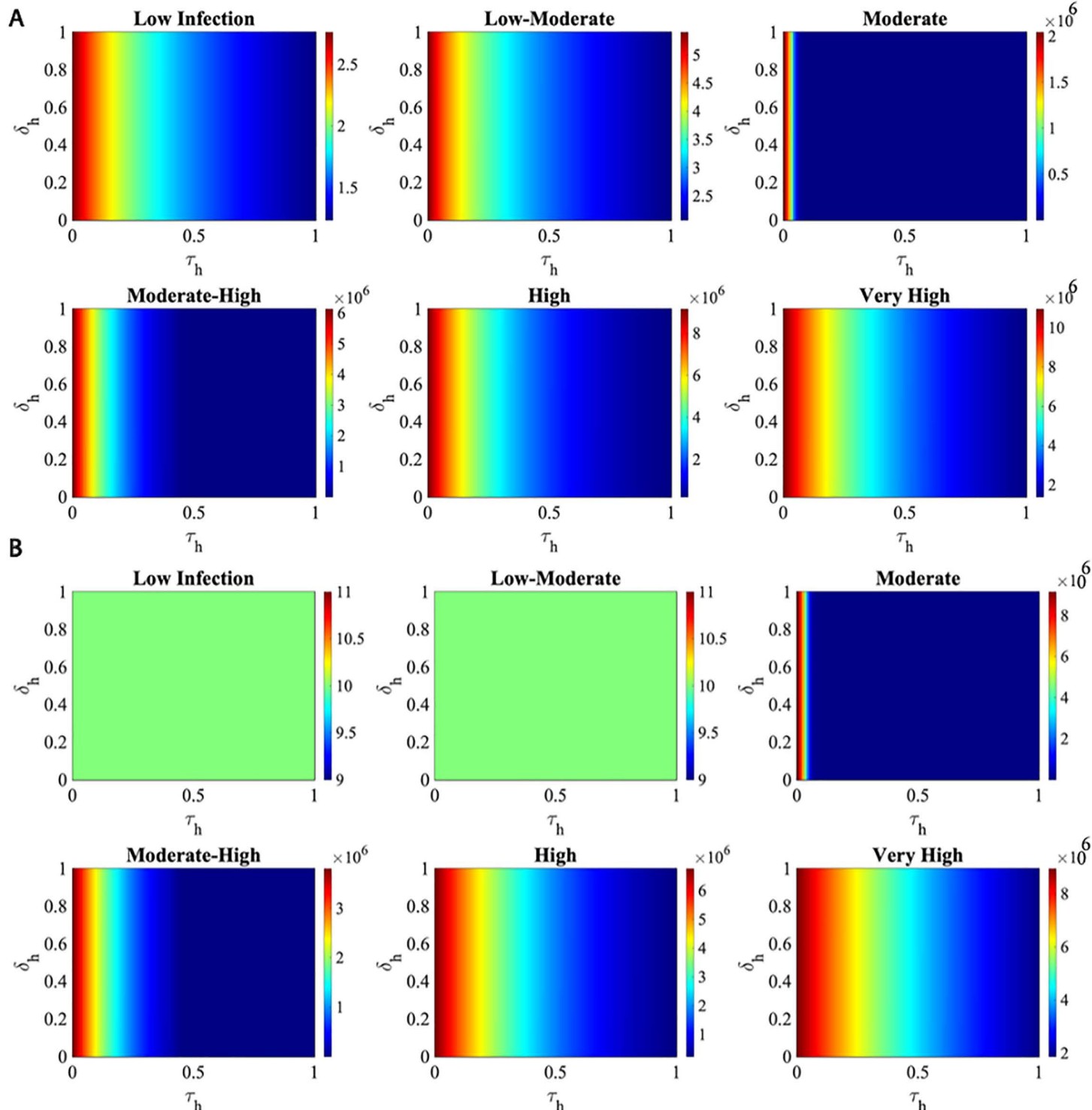

**Fig 8. Heatmaps of peak simulated infection levels in humans and vectors for the six infection risk scenarios with differing treatment and recovery rates.**

is back, represented by the vertical line pattern, similar to Fig 8A. This could indicate a threshold effect, where the parameter $\tau_h$ has an effect only after a certain level of severity is reached. In both panels, the colors don't really change much in the vertical direction, indicating that the parameter $\tau_h$ is the driving force, where the parameter $\delta_h$ has little to no effect in the parameter space.

Fig 9 examine how varying levels of treatment impact the spread of infection in both humans and mosquitoes across six scenarios, ranging from low to very high infection risk. Infection levels ($I_h$ and $I_v$) escalate rapidly, with the red (Very High) and magenta (High) for the first 40 weeks. We examined three treatment regimens: *no treatment* (treatment rate, $\tau_h = 0$ and recovery rate, $\delta_h = 0$) *moderate treatment* ($\tau_h = 0.25$, $\delta_h = 0.5$) and *strong treatment* ($\tau_h = 0.5$, $\delta_h = 0.9$). Without treatment, infections rapidly increase and reach high peaks, especially in the higher-risk scenarios. Introducing moderate and high treatments significantly reduces the number of infections and shortens outbreak duration, with the greatest impact observed in lower-risk scenarios. Under intense treatment, infection peaks are much smaller, and outbreaks progress more slowly across all scenarios, demonstrating the potential to control even very high transmission levels. These results highlight that both the extent of treatment coverage and its effectiveness play critical roles in managing disease spread in human and mosquito populations.

These simulation results identify the key epidemiological drivers of dengue dynamics and provide a quantitative basis for fitting the model to observed data. In the calibration process, the epidemiological parameters $v = [\tau_h, \beta_h, \gamma_h, \beta_v, b, \rho]$

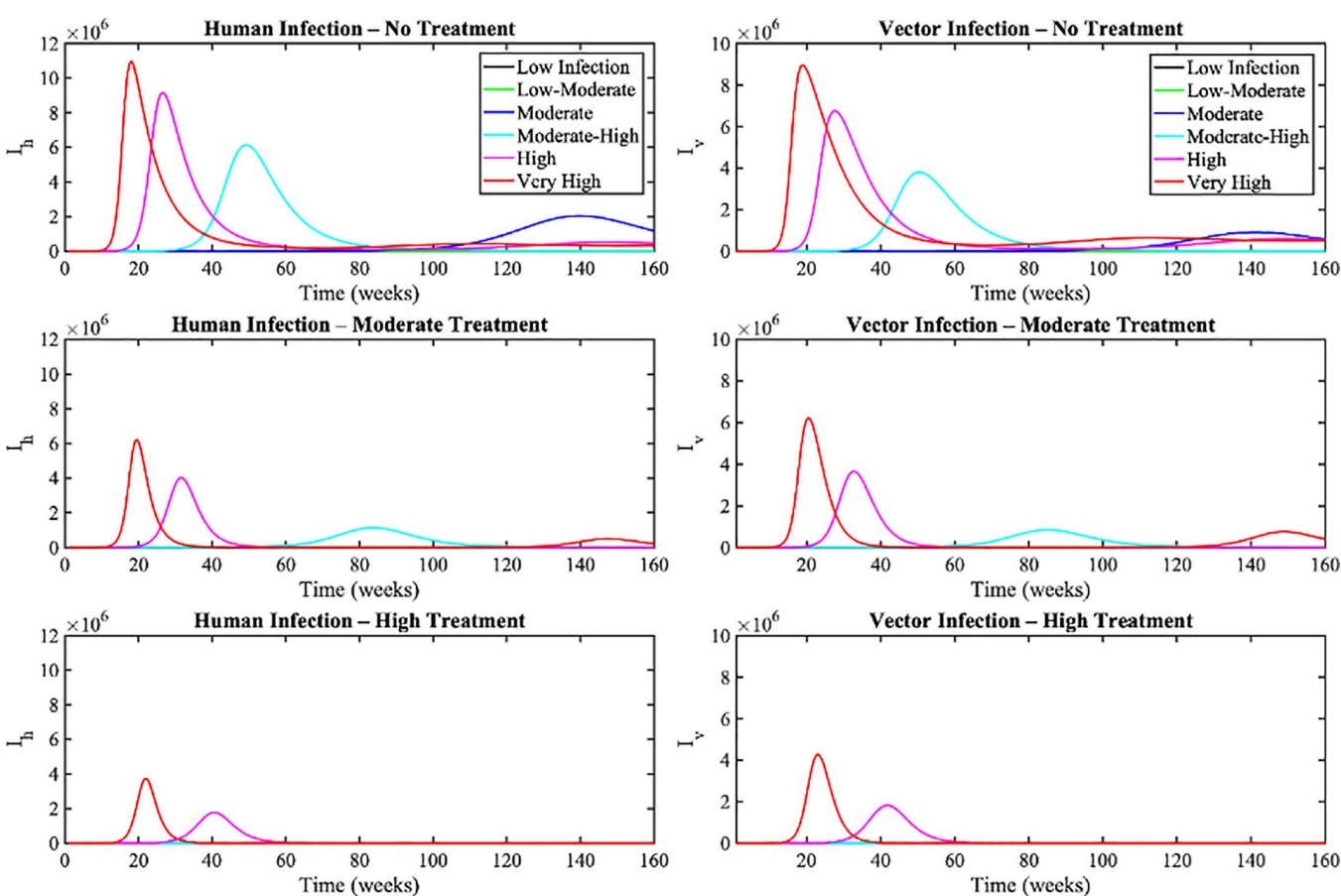

**Fig 9. Impact of varying treatment levels on infection dynamics in human and vector populations across six risk scenarios.**

were estimated, while the remaining parameters were held fixed as specified in Table 2 and S1 Table. This approach ensures that the model is grounded in the most influential parameters while maintaining consistency with the established baseline values.

## 4. Model calibration

In this section, we calibrated the proposed model using weekly dengue incidence data across the eight divisions and the Dhaka Metropolitan, offering valuable insights for informing proactive public health interventions.

Suppose we are given the data $\{(t_1, y_1),\ (t_2, y_2),\ \ldots,\ (t_n, y_n)\}$, where $y_n$ is the *n-th* observed data point, and $n$ is the total number of data points. We aim to fit $\rho I(t),$ where $I(t)$ is the model-predicted incidence and and $\rho$ is a "reporting rate," using the model parameter $v$. Then the goodness of fit is quantified using the sum of squared errors (SSE):

$$L := SSE(v) = \sum_{j=1}^{n} \|y_j - \rho I(t_j)\|^2$$

We estimate the model parameter $\hat{v}$ of the model parameter $v$ by minimizing SSE using a least-squares approach. The reporting rate ($\rho$) is defined as the proportion of actual cases in a population that are reported by the surveillance system ($\rho$ indicates complete reporting, while $\rho < 1$ signifies underreporting). The model is simulated using MATLAB's ode45, and parameter estimation is performed with fmincon to minimize SSE.

### 4.1. Temperature dependent biting rate

The vector-host transmission is a cycle between the vector (female mosquito) and the host (human). When infected mosquitoes bite humans, the humans become infectious after an incubation period and can subsequently transmit the virus back to mosquitoes. Although relative humidity exhibits the strongest contemporaneous correlation with dengue cases (see S1 and S2 Figs), this does not imply a greater mechanistic role. Humidity primarily influences mosquito survivorship and breeding site persistence, which tend to align closely with seasonal rainfall patterns. In contrast, temperature affects transmission through the biting rate, development rate, and extrinsic incubation period. These effects are nonlinear and often lagged, resulting in weaker raw correlations with weekly incidence. Using the temperature-dependent biting rate $b(T)$, therefore, captures the mechanistic influence of temperature more accurately than raw temperature values. Supplementary analysis also shows that $b(T)$ correlates more strongly with incidence than raw temperature, supporting its use in the model. Additionally, as described by [13], the mosquito biting rate is a critical determinant of dengue transmission and is temperature dependent. Recent studies using fractional-order models have also further highlighted that biting rate is one of the most sensitive parameters influencing dengue transmission dynamics, emphasizing the importance of accurately capturing its temperature dependence [66]. To capture this effect, we use average weekly temperature data to calculate the corresponding biting rate using the formula:

$$b(T) = \begin{cases} 0.0943 + 0.0043T & 21^{\circ}\text{C} < T < 32^{\circ}\text{C} \\ 0, & otherwsie \end{cases}.$$

as described in [66], where $T$ represents the weekly average temperature. In this study, we update this equation with the weekly average temperatures across the studied locations. The resulting function, $b(T)$, thus provides a temperature-dependent measure of the mosquito biting rate.

Fig 10 illustrates the variability of temperature data across the eight divisions and the Dhaka Metropolitan area, focusing on the years 2024–2025. Fig 10A shows temperature variability across the divisions for 2024 and 2025. Fig 10B presents the weekly average temperature ($^{\circ}C$) for the Dhaka Metropolitan area along with the corresponding mosquito biting rate function, $b(T)$.

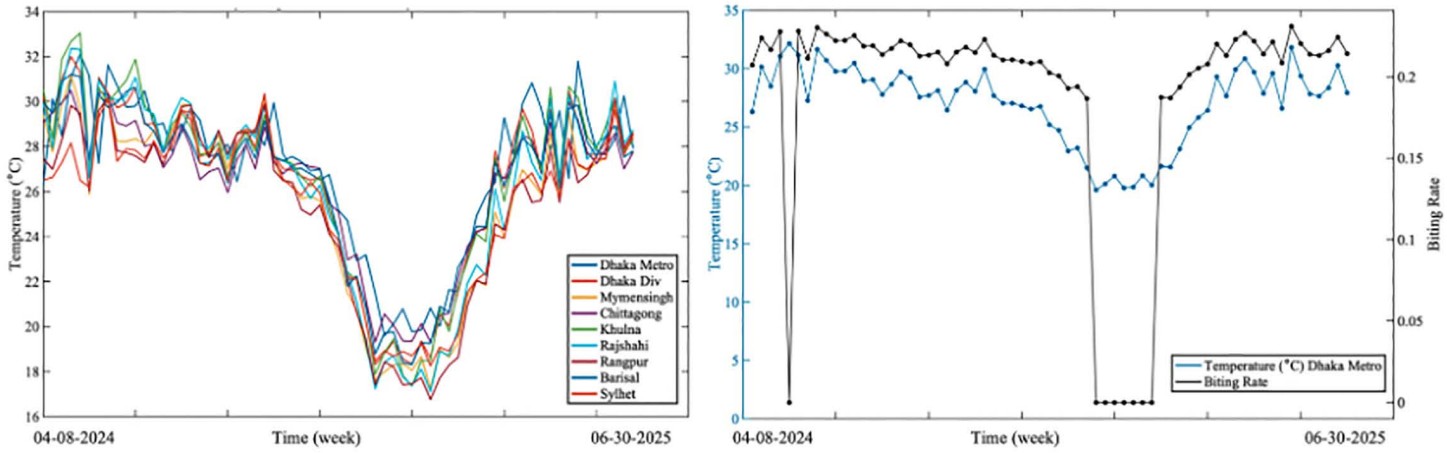

**Fig 10. Temperature variability across locations in Bangladesh and the temperature-dependent mosquito biting rate.**

### 4.2. Calibration data and targets

We calibrated the model using weekly reported dengue case data across all eight divisions of Bangladesh: Barishal, Chittagong, Dhaka, Khulna, Rajshahi, Rangpur, Mymensingh, Sylhet, and the Dhaka Metropolitan area, spanning April 13, 2024, to August 23, 2025. The epidemiological parameters $v = [\tau_h, \beta_h, \gamma_h, \beta_v, b, \rho]$ were estimated, while the remaining parameters were held fixed as specified in Table 2 and S1 Table. For the simulation, the initial susceptible population was set to the actual population size provided in S1 Table. The initial number of infectious individuals was defined as $I_{h0} = \frac{1}{\rho} I_h^0$ ,where $I_h^0$ corresponds to the reported cases from the epidemiological week of April 13, 2024 and $\rho = \frac{1}{10}$. In the Rajshahi and Rangpur Divisions, where no cases were reported, we assumed a single case to initialize the model. We also provided reported basic reproduction number $(R_0)$ across different countries $(R_0)$ across different countries (see S2 Table).

### 4.3. Model fitting and forecasting

Following model calibration, we applied the estimated parameters to fit the model to the reported dengue incidence data and to generate short-term forecasts. Fig 11 displays reported dengue cases (red), model fitting (blue), and forecasted cases (magenta). Dashed blue lines indicate the end of the fitting period (April 13, 2024, to July 05, 2025, EW 15, 2024 to EW 27, 2025), while the solid black line marks the beginning of the forecasting period (July 12 to August 23, 2025, EW 28 to EW 34, 2025). This clear separation avoids data overlap, allowing forecasts for an independent period while maintaining alignment with epidemiological weeks. Parameter bounds used were: lower [0.0001, 0.0001, 0.0001, 0.0001, 0.0001] and upper [1, 1, 3, 1, 4, 0.04] for $[\tau_h, \beta_h, \gamma_h, \beta_v, b, \rho]$. The resulting fitted parameters from July 12, 2025, to August 23, 2025 dataset data are given in S3 and S4 Tables.

Reported dengue cases (red) are shown together with the model fit (blue) and short-term forecasts (magenta). Vertical dashed blue lines indicate the end of the calibration period, and the solid black vertical line marks the beginning of the forecasting window. Shaded bands denote predictive uncertainty, with intervals corresponding to 95% (yellow), 90% (light orange), 80% (medium blue), and 50% (green). Forecast performance varies across divisions, with greater uncertainty in regions such as Rangpur and Sylhet where reported incidence is low.

### 4.4. Model validation

To evaluate the validity of the calibrated model, we compared model-estimated dengue cases with reported observations across nine administrative divisions during the fitting period. Fig 12 illustrates the agreement between estimated

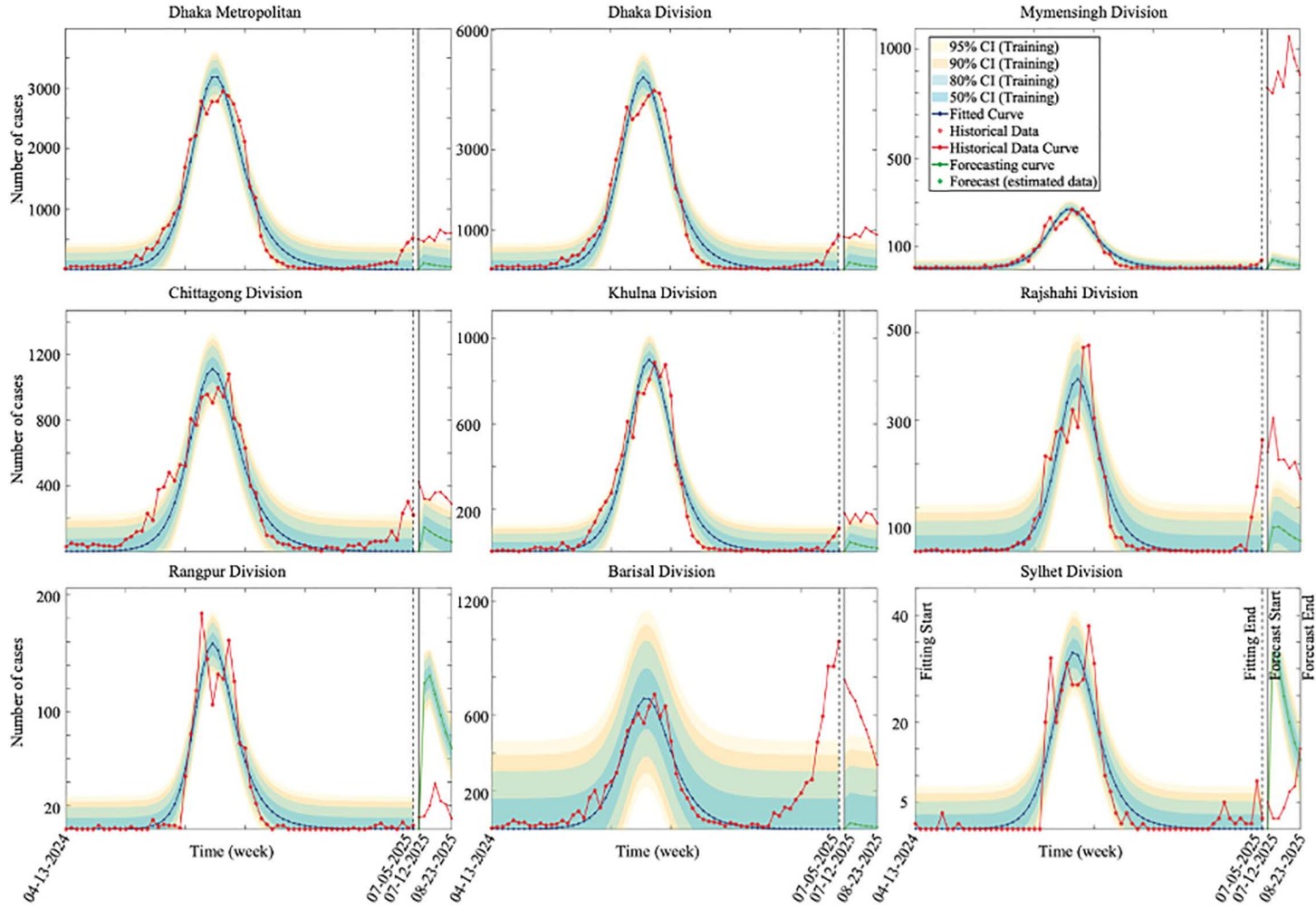

**Fig 11. Fitting and short-term forecasting of probable dengue cases across locations in Bangladesh.**

and observed cases using scatter plots and corresponding coefficients of determination ($R^2$). Overall, the results indicate strong model performance, with most divisions exhibiting high correlations, reflecting the model's ability to capture regional dengue dynamics accurately. In contrast, Barisal Division shows a substantially lower $R^2$ value, suggesting greater variability and potential limitations related to data quality, reporting, or local transmission heterogeneity. Despite this regional discrepancy, the validation results demonstrate that the model provides a reliable representation of dengue incidence across the majority of regions, supporting its use for inference and short-term forecasting.

Scatter plots show the relationship between model-estimated and reported cases for each division, with the red dashed line indicating perfect agreement. The coefficient of determination ($R^2$) quantifies the strength of the association between estimated and observed cases. Most divisions exhibit strong agreement ($R^2 > 0.8$), indicating good model performance, whereas Barisal Division shows a weaker correlation ($R^2 = 0.20$), suggesting greater regional variability in model fit.

### 4.5. Treatment effects on transmission

To assess the role of treatment in reducing dengue transmission, we examined how variations in treatment-related parameters influence the effective reproduction number across regions. The results indicate that increased treatment coverage

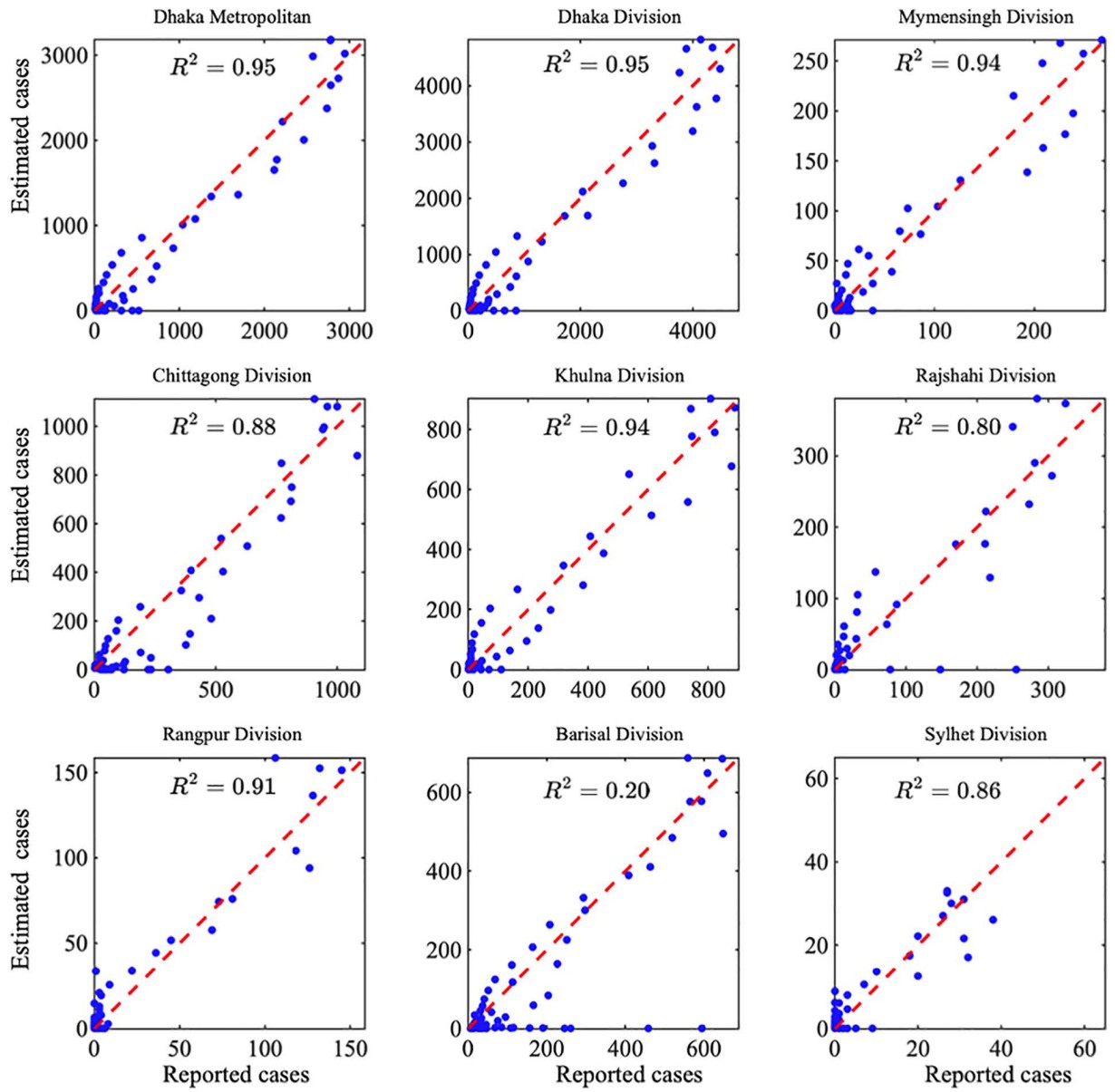

**Fig 12. Comparison of estimated and reported dengue cases during the fitting period across nine administrative regions of Bangladesh.**

is generally associated with reduced transmission potential, reflecting the role of timely case management in shortening infectious periods and lowering onward spread. However, the relationship between treatment intensity and transmission reduction is not uniform across regions, highlighting the influence of complementary factors such as recovery rates and overall healthcare effectiveness. In particular, high treatment levels alone may be insufficient to suppress transmission when treatment recovery is limited, underscoring the importance of supportive care and integrated intervention strategies. Overall, these findings demonstrate that treatment can substantially mitigate dengue transmission when combined with effective recovery and healthcare capacity, reinforcing the need for coordinated clinical and public health interventions.

Fig 13 examines treatment levels and their impact on dengue transmission across Bangladesh. Fig 13A Bar plot of the treatment parameter ($\tau_h$) across eight divisions, including Dhaka Metropolitan, showing higher treatment levels in Dhaka Metropolitan, Chittagong, and Rajshahi and lower levels in Sylhet. Fig 13B Scatter plot of treatment ($\tau_h$) versus reproduction number ($R_c$), with a fitted trend line illustrating that higher treatment generally corresponds to lower $R_c$, and improved model fit. Rajshahi Division is highlighted as an exception, where high treatment alone did not reduce $R_c$, due to minimal recovery rates S4 Table, emphasizing the importance of supportive care alongside treatment interventions.

## 5. Discussion

Dengue fever remains a significant public health challenge in Bangladesh, with increasing morbidity and mortality reported in recent years due to the severity of the disease. Endemic for over a decade, dengue is now widespread across the country, particularly affecting urban and semi-urban populations, and is particularly visible at the division level (see Fig 1). The disease thrives in tropical and subtropical climates, making Bangladesh highly susceptible to recurrent outbreaks. Although dengue was initially reported in Bangladesh during the 1960s, the first officially recorded dengue outbreak in the country occurred in the year 2000, marking the beginning of a persistent public health concern. Since then, the frequency and intensity of dengue outbreaks have escalated, placing a considerable burden on the healthcare system. The transmission is facilitated by the *Aedes aegypti* mosquito, which breeds in stagnant water sources commonly found

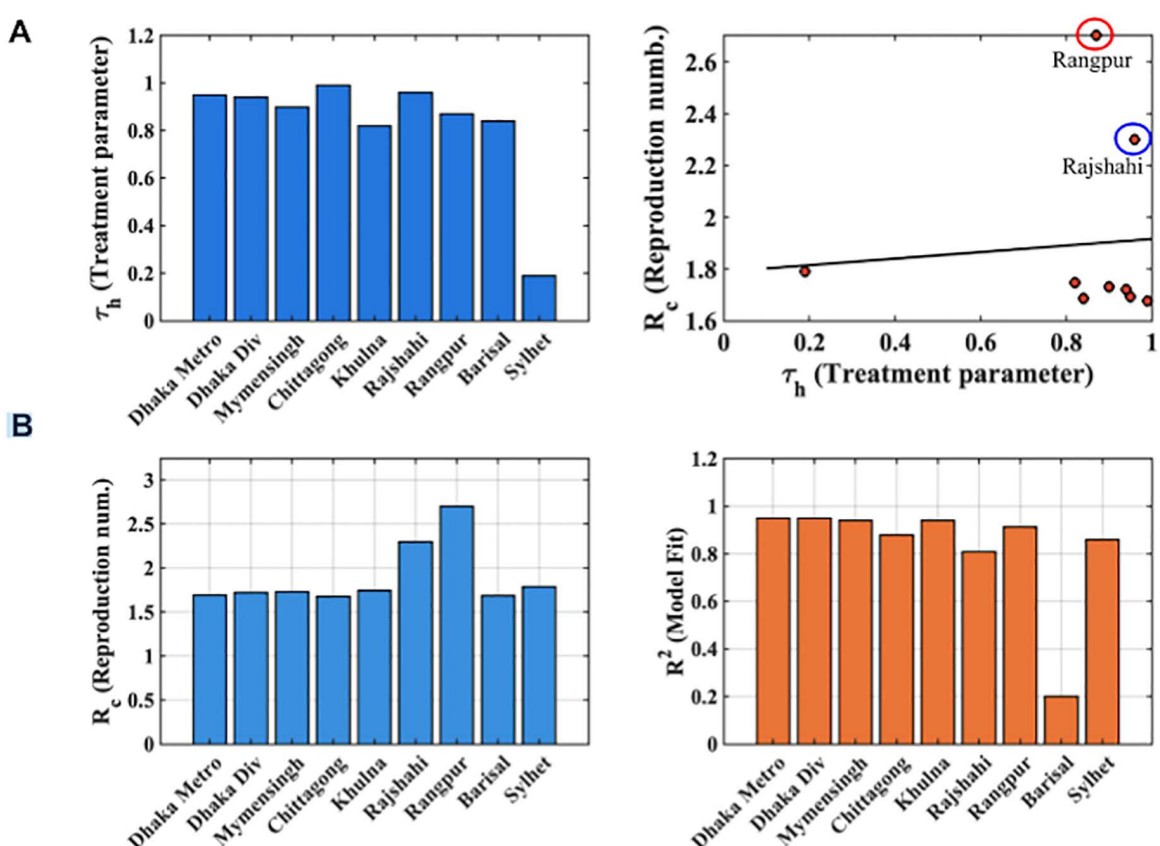

**Fig 13. Impact of treatment parameters on dengue transmission across Bangladesh.**

in densely populated areas. The growing urbanization, climate variability, and inadequate vector control measures have contributed to the sustained endemicity and periodic surges in dengue cases across the country.

In this study, we constructed a temperature-dependent human-mosquito transmission model with treatment to investigate the dynamics of dengue transmission (see Figs 6 and 7) and control at the division level in Bangladesh. As highlighted in S5 Table, previous studies emphasize supportive care but lack of quantitative analysis of treatment effects, demonstrating the necessity of explicitly including treatment in our model. While no specific antiviral cure exists for dengue, current treatment focuses on supportive care, such as intravenous fluids and management of fever and complications (plasma exchange), which can influence the recovery rate. The emphasis in this paper is on medical treatment of infected individuals and how its accessibility and efficacy affect outbreak dynamics. By integrating both treatment and temperature effects, our study fills this gap and provides a rigorous framework for understanding dengue dynamics and guiding control strategies at the provincial level in Bangladesh. Our model revealed two equilibrium points: a disease-free equilibrium and an endemic equilibrium. The basic reproduction number ($R_c$) was calculated using the next-generation matrix approach. The values of the basic reproduction number ($R_c$) and biological characteristics of the suggested model has been established using available data, as shown in Table 2 and S6 Table. Furthermore, using the eigenvalue equation, the global stability of SITRS is being studied. We discovered that the threshold quantity, also known as the basic reproduction number, determines the system's equilibrium states and their stability. The results indicate that if the basic reproduction number ($R_c$) is less than 1, naturally dengue fever fades out, implying that dengue dies. If $R_c > 1$, dengue illness persists in the population.

In addition, we performed the analytical and numerical analysis for equilibriums that have both disease free and endemic equilibrium. In order to assess the chance of infection spreading from humans to mosquitoes, we applied the least-square fitting strategy to calibrate dengue incidence data with the model. The numerical simulations were carried out to determine the impact of the birth and mortality rate of human population ($\mu_h$), treatment rate ($\tau_h$), natural recovery rate ($\gamma_h$) and recovery rate from treatment ($\delta_h$) on dengue prevalence. Our numerical simulations (see Figs 8 and 9) demonstrated that enhancing treatment, such as supportive care, can reduce the peak number of infections during outbreaks, underscoring the critical role treatment plays in controlling dengue in Bangladesh. The findings show that an increase in the percentage of treatment rate and rate of immunity decline has an enormous effect on the spread of the infection among those affected. The implications of these findings for choosing appropriate preventive measures are significant. On the other hand, when the treatment rate ($\tau_h$) decreases, there are more individuals with illnesses.

In conclusion, this study aimed to enhance understanding of dengue transmission dynamics in Bangladesh by emphasizing the role of effective treatment strategies in improving the human recovery rate. Through a comprehensive mathematical model, we demonstrated that the basic reproduction number is influenced by several key parameters, including the human-to-mosquito ($\beta_h$) and mosquito-to-human ($\beta_v$) transmission rates, the treatment rate of the disease ($\tau_h$), mortality in humans and mosquitoes ($\mu_h$ and $\mu_v$), the mosquito biting rate ($b$), the rate of immunity loss ($\theta_h$) and the human recovery rate ($\gamma_h$). Our sensitivity analysis revealed that the transmission rates ($\beta_h$ and $\beta_v$) and the biting rate ($b$) are the most influential factors driving dengue transmission. These findings underscore the critical importance of targeted vector control strategies, public awareness efforts to reduce mosquito-human contact, and the provision of timely and effective medical treatment. By identifying and quantifying the impact of key transmission parameters, our model was also calibrated using epidemiological case data from the Directorate General of Health Services (DGHS) across Bangladesh's eight administrative divisions, as well as the Dhaka Metropolitan area, with the temperature variability data at the division level (see Fig 10). The data fitting (Fig 11) and the comparison of estimated versus observed cases (Fig 12) revealed distinct peak magnitudes across the divisions. However, the timing of the peaks is broadly similar, reflecting strong seasonality and the influence of temperature. Furthermore, by incorporating temperature data to account for climatic influences on mosquito biting rates, the model provides short-term forecasts for 2025 in Fig 11, highlighting critical thresholds and guiding potential intervention strategies. These results underscore that while increased treatment

coverage can substantially reduce dengue transmission in most regions, as shown in Fig 12, its effectiveness depends critically on recovery from supportive care, highlighting the need for integrated treatment (see Fig 13) and supportive care strategies across Bangladesh.

Future research should focus on public education and the promotion of preventive practices to combat dengue transmission. Forthcoming studies may investigate innovative strategies to reduce mosquito-to-human transmission, including the use of environmentally sustainable and natural treatment options that minimize vector populations without harmful side effects. Moreover, identifying and evaluating the most effective policy interventions—both at the local and national levels—remains a critical area of inquiry. This includes assessing the impact of existing public health policies and developing evidence-based guidelines tailored to the specific socio-environmental conditions of endemic regions. Additionally, enhancing community engagement through targeted outreach campaigns via social media, television, radio, and other mass communication platforms can play a transformative role in fostering behaviour change. These campaigns should aim to raise awareness about the importance of regular environmental cleaning, elimination of mosquito breeding sites, and early health-seeking behaviour. By integrating scientific research with community-based approaches and communication strategies, future efforts can significantly strengthen dengue prevention and control.

## Supporting information

**S1 Fig. Reported dengue cases (Panel A), deaths (Panel B), and case fatality rate (CFR, Panel C) in Bangladesh for the years 2018, 2019, 2021, 2022, 2023, 2024, and 2025 (as of Sept. 7, 2025).**
(TIF)

**S2 Fig. Correlation of dengue cases with key climate variables across Bangladesh.**
(TIF)

**S1 Table. Administrative division populations of Bangladesh, 2022 census results.**
(PDF)

**S2 Table. Reported basic reproduction numbers ($R_0$) cross different countries.**
(PDF)

**S3 Table. Sensitivity indices of $R_c$ to the parameters for the model.**
(PDF)

**S4 Table. Estimated parameters, associated errors, $R^2$ and $R_c$.**
(PDF)

**S5 Table. Estimated reproduction numbers $R_c$ for different divisions of Bangladesh.**
(PDF)

**S6 Table. Summary of literature on dengue in Bangladesh, highlighting treatment- related aspects, biting behaviour, temperature influences, and references.**
(PDF)

## Author contributions

**Conceptualization:** Md Mafizer Rahman, Haridas K. Das, Sazia Khatun Tithi, Md Tonmoy Ul Hasan, Moyuri Khatun, Md Abdul Kuddus.

**Data curation:** Haridas K. Das, Md Abdul Kuddus.

**Formal analysis:** Md Mafizer Rahman, Haridas K. Das.

**Funding acquisition:** Haridas K. Das, Md Abdul Kuddus.

**Investigation:** Md Mafizer Rahman, Haridas K. Das, Md Abdul Kuddus.

**Methodology:** Md Mafizer Rahman, Haridas K. Das, Md Abdul Kuddus.

**Project administration:** Haridas K. Das, Md Abdul Kuddus.

**Resources:** Haridas K. Das, Md Abdul Kuddus.

**Software:** Md Mafizer Rahman, Haridas K. Das.

**Supervision:** Haridas K. Das, Md Abdul Kuddus.

**Validation:** Md Mafizer Rahman, Haridas K. Das, Sazia Khatun Tithi, Md Tonmoy Ul Hasan, Moyuri Khatun, Md Abdul Kuddus.

**Visualization:** Md Mafizer Rahman, Haridas K. Das, Sazia Khatun Tithi, Md Tonmoy Ul Hasan, Moyuri Khatun, Md Abdul Kuddus.

**Writing – original draft:** Md Mafizer Rahman, Haridas K. Das.

**Writing – review & editing:** Md Mafizer Rahman, Haridas K. Das, Sazia Khatun Tithi, Md Tonmoy Ul Hasan, Moyuri Khatun, Md Abdul Kuddus.

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
