## [Decision Letter · Decision Letter 0]

26 Dec 2025

PONE-D-25-62939Modeling Treatment and Temperature Effects on Dengue Transmission at the Division Level in BangladeshPLOS One

Dear Dr. Kuddus,

Thank you for submitting your manuscript to PLOS ONE. After careful consideration, we feel that it has merit but does not fully meet PLOS ONE’s publication criteria as it currently stands. Therefore, we invite you to submit a revised version of the manuscript that addresses the points raised during the review process.

We look forward to receiving your revised manuscript.

Kind regards,

Md. Kamrujjaman, Ph.D

Academic Editor

PLOS One

Journal Requirements:

3. Thank you for uploading your study's underlying data set. Unfortunately, the repository you have noted in your Data Availability statement does not qualify as an acceptable data repository according to PLOS's standards.

4. Please ensure that you refer to Figure 9 in your text as, if accepted, production will need this reference to link the reader to the figure.

5. We note that Figure 1 in your submission contain map images which may be copyrighted. All PLOS content is published under the Creative Commons Attribution License (CC BY 4.0), which means that the manuscript, images, and Supporting Information files will be freely available online, and any third party is permitted to access, download, copy, distribute, and use these materials in any way, even commercially, with proper attribution. For these reasons, we cannot publish previously copyrighted maps or satellite images created using proprietary data, such as Google software (Google Maps, Street View, and Earth). For more information, see our copyright guidelines: http://journals.plos.org/plosone/s/licenses-and-copyright.

6. We note you have included a table to which you do not refer in the text of your manuscript. Please ensure that you refer to Table 1 in your text; if accepted, production will need this reference to link the reader to the Table.

Reviewer's Responses to Questions

**Comments to the Author**

1. Is the manuscript technically sound, and do the data support the conclusions?

Reviewer #1: Yes

Reviewer #2: Yes

2. Has the statistical analysis been performed appropriately and rigorously? 

Reviewer #1: Yes

Reviewer #2: Yes

3. Have the authors made all data underlying the findings in their manuscript fully available?

Reviewer #1: Yes

Reviewer #2: Yes

4. Is the manuscript presented in an intelligible fashion and written in standard English?

Reviewer #1: Yes

Reviewer #2: Yes

5. Review Comments to the Author

Reviewer #1: The authors did a good job, but I have few comments to make:

1. Add proper reference to the following lines in the Introduction Section:

The virus persists in the human bloodstream for two to seven days………

Globally, dengue imposes a heavy public health burden, with an estimated 400 million infections annually, ……………….

Between 1990 and 2023, reported cases increased by 88%, though……..

In 2024, cases declined to 101,214 with 575 deaths, while…………..

Importantly, temperature shows the strongest correlation with dengue cases among the climate variables ……..

2. In page 6, mention the expression of N_h in the disease free equilibrium (DFE) like N_v.

3. Modify the structure of column 6 and column 7 in Table 2. Put the values in column 6 and units in column 7, do not write together.

4. In column 4 of Table 2, the values of Rc should be based on column 2, not column 3.

5. In section 2.9, write a clear and rigorous explanation of the sensitivity index. For example, the sensitivity index of Rc with respect to the parameter beta_h is 1, what does it mean ?

6. What initial conditions were used for each compartment ?

7. How the parameter values were chosen for different scenarios (low, low-moderate, moderate, high-moderate, very high) in page 17 ? Were these values just assumed or is there any significance ?

8. In the very high infection scenario, b =1.25. Can b be greater than 1 ? If yes, why ?

9. The paper mentions that parameter values were estimated using MATLAB’s fmincon. What are the confidence intervals for these estimates, and how were these parameters initialized during the optimization process?

10. In page 10, the paper mentions Routh–Hurwitz criterion, but neither stated it nor gave any references.

Reviewer #2: Minor Comments:

1. The manuscript contains several spelling inconsistencies (e.g. suppohomorting) that should be carefully reviewed and corrected throughout.

2. The reference numbering does not appear to follow a consistent order throughout the manuscript. Please review and revise the placement of references to ensure proper sequencing.

3. A careful check of the reference list is recommended, as certain references seem to appear repeatedly.

4. The parameter $\tau_h$ is inconsistently described as a specified time interval, a disease progression rate, and a treatment rate in different sections. Clarification and revision are requested to ensure consistency.

5. Please check the formatting of columns 6 and 7 of table 2, as they appear to be merged.

6. Please carefully check the x-tick labels in Fig. 2, Panel B, as they may require correction.

7.Please carefully revise the theorem (Theorem 3) statement to ensure consistency between the stability condition and the biological interpretation.

Major Comments:

1.Given that Fig. 8 depicts scenarios for human infections $(I_h)$. Clarification is requested on how conclusions regarding infection dynamics in both human and mosquito populations were drawn, and what analyses were conducted to assess the role of mosquito control under epidemic conditions.

2. Could you please clarify how the analysis presented in Fig. 9 supports conclusions regarding infection dynamics in the mosquito population? While treatment may indirectly reduce mosquito infections by lowering human infectiousness, it would be helpful to explain how such conclusions were drawn in the absence of explicit vector control interventions or mosquito-targeted measures in the model.

3. The temperature-dependent biting rate function b(T) is directly adopted from reference [43]. Could you please clarify the rationale for applying this formulation in the Bangladesh context? In particular, clarification is requested regarding the suitability of the adopted formulation for Bangladesh-specific conditions and whether any validation or sensitivity analyses were performed.

4.Figure 5, Panel A is described as using the baseline parameter values listed in Table 2; however, Table 2 reports a basic reproduction number $R_c=3.0456$ for these parameter values, whereas Figure 5 indicates $R_c<1$. Could you please clarify this apparent discrepancy and explain how the condition $R_c<1$ was obtained in Figure 5?

5.The figure description (Fig. 6) appears to be inconsistent with the content shown in the figure. Clarification or revision would be appreciated to ensure consistency between the figure and its description.

6.While sensitivity analyses of epidemiological parameters in human and vector classes are presented, the inclusion of an LHS–PRCC approach could further account for parameter uncertainty.

7. Please consider placing figures within their corresponding sections in the main manuscript and integrating the correlation figures from the supporting file where appropriate.

8.The manuscript would benefit from engagement with additional literature on dengue transmission, treatment-based intervention modeling, and uncertainty quantification, as well as broader infectious disease modelling studies on host–vector dynamics, threshold analysis, and control strategies.

a)Cost-effectiveness of dengue control strategies in Bangladesh: An optimal control and ACER-ICER analysis. https://doi.org/10.1016/j.actatropica.2025.107587

b)Optimal control analysis of COVID-19 transmission model with physical distance and treatment. http://dx.doi.org/10.26855/abr.2022.12.001.

c)Analysis of a Data‐Driven Vector‐Borne Dengue Transmission Model for a Tropical Environment in Bangladesh. https://doi.org/10.1155/2024/2959770

d)Stochastic analysis of Mpox epidemiology with vaccination strategies and environmental persistence. https://doi.org/10.1038/s41598-025-28135-x

6. PLOS authors have the option to publish the peer review history of their article (what does this mean?). If published, this will include your full peer review and any attached files.

Reviewer #1: No

Reviewer #2: No

---

## [Author Response · Author response to Decision Letter 1]

8 Feb 2026

Please find the attached "response to editor and reviewer comments file".

---

## [Decision Letter · Decision Letter 1]

19 Feb 2026

PONE-D-25-62939R1Modeling Treatment and Temperature Effects on Dengue Transmission at the Division Level in BangladeshPLOS One

Dear Dr. Kuddus,

Thank you for submitting your manuscript to PLOS ONE. After careful consideration, we feel that it has merit but does not fully meet PLOS ONE’s publication criteria as it currently stands. Therefore, we invite you to submit a revised version of the manuscript that addresses the points raised during the review process.

We look forward to receiving your revised manuscript.

Kind regards,

Md. Kamrujjaman, Ph.D

Academic Editor

PLOS One

Journal Requirements:

Reviewers' comments:

Reviewer's Responses to Questions

**Comments to the Author**

1. If the authors have adequately addressed your comments raised in a previous round of review and you feel that this manuscript is now acceptable for publication, you may indicate that here to bypass the “Comments to the Author” section, enter your conflict of interest statement in the “Confidential to Editor” section, and submit your "Accept" recommendation.

Reviewer #1: All comments have been addressed

Reviewer #2: (No Response)

2. Is the manuscript technically sound, and do the data support the conclusions?

Reviewer #1: Yes

Reviewer #2: (No Response)

3. Has the statistical analysis been performed appropriately and rigorously? 

Reviewer #1: N/A

Reviewer #2: (No Response)

4. Have the authors made all data underlying the findings in their manuscript fully available?

Reviewer #1: Yes

Reviewer #2: (No Response)

5. Is the manuscript presented in an intelligible fashion and written in standard English?

Reviewer #1: Yes

Reviewer #2: (No Response)

6. Review Comments to the Author

Reviewer #1: (No Response)

Reviewer #2: (No Response)

7. PLOS authors have the option to publish the peer review history of their article (what does this mean?). If published, this will include your full peer review and any attached files.

Reviewer #1: No

Reviewer #2: No

---

## [Author Response · Author response to Decision Letter 2]

2 Apr 2026

Please find attached the document titled ‘Response to Editor and Reviewer comments’ for your review.

---

## [Decision Letter · Decision Letter 2]

13 Apr 2026

Modeling Treatment and Temperature Effects on Dengue Transmission at the Division Level in Bangladesh

PONE-D-25-62939R2

Dear Dr. Kuddus,

We’re pleased to inform you that your manuscript has been judged scientifically suitable for publication and will be formally accepted for publication once it meets all outstanding technical requirements.

Kind regards,

Md. Kamrujjaman, Ph.D

Academic Editor

PLOS One

Additional Editor Comments (optional):

Reviewers' comments:

Reviewer's Responses to Questions

**Comments to the Author**

1. If the authors have adequately addressed your comments raised in a previous round of review and you feel that this manuscript is now acceptable for publication, you may indicate that here to bypass the “Comments to the Author” section, enter your conflict of interest statement in the “Confidential to Editor” section, and submit your "Accept" recommendation.

Reviewer #2: (No Response)

2. Is the manuscript technically sound, and do the data support the conclusions?

Reviewer #2: (No Response)

3. Has the statistical analysis been performed appropriately and rigorously? 

Reviewer #2: (No Response)

4. Have the authors made all data underlying the findings in their manuscript fully available?

Reviewer #2: (No Response)

5. Is the manuscript presented in an intelligible fashion and written in standard English?

Reviewer #2: (No Response)

6. Review Comments to the Author

Reviewer #2: (No Response)

7. PLOS authors have the option to publish the peer review history of their article (what does this mean?). If published, this will include your full peer review and any attached files.

Reviewer #2: No

---

## [Editor Report · Acceptance letter]

PONE-D-25-62939R2

PLOS One

Dear Dr. Kuddus,

I'm pleased to inform you that your manuscript has been deemed suitable for publication in PLOS One. Congratulations! Your manuscript is now being handed over to our production team.

Kind regards,

on behalf of

Dr. Md. Kamrujjaman

Academic Editor

PLOS One